# LIFTING ARCHITECTURAL CONSTRAINTS OF INJECTIVE FLOWS

**Peter Sorrenson,**[*] **Felix Draxler,**[*] **Armand Rousselot,**
**Sander Hummerich, Lea Zimmermann & Ullrich Köthe**
Computer Vision and Learning Lab
Heidelberg University
`firstname.lastname@iwr.uni-heidelberg.de`

## ABSTRACT

Normalizing Flows explicitly maximize a full-dimensional likelihood on the training data. However, real data is typically only supported on a lower-dimensional manifold leading the model to expend significant compute on modeling noise. Injective Flows fix this by jointly learning a manifold and the distribution on it. So far, they have been limited by restrictive architectures and/or high computational cost. We lift both constraints by a new efficient estimator for the maximum likelihood loss, compatible with free-form bottleneck architectures. We further show that naively learning both the data manifold and the distribution on it can lead to divergent solutions, and use this insight to motivate a stable maximum likelihood training objective. We perform extensive experiments on toy, tabular and image data, demonstrating the competitive performance of the resulting model.

## 1 INTRODUCTION

Generative modeling is one of the most important tasks in machine learning, having numerous applications across vision (Rombach et al., 2022), language modeling (Brown et al., 2020), science (Ardizzone et al., 2018; Radev et al., 2022) and beyond. One of the best-motivated approaches to generative modeling is maximum likelihood training, due to its favorable statistical properties (Hastie et al., 2009). In the continuous setting, exact maximum likelihood training is most commonly achieved by normalizing flows (Rezende & Mohamed, 2015; Dinh et al., 2015; Kobyzev et al., 2021) which parameterize an exactly invertible function with a tractable change of variables (log-determinant term). This generally introduces a trade-off between model expressivity and computational cost, where the cheapest networks to train and sample from, such as coupling block architectures, require very specifically constructed functions which may limit expressivity (Draxler et al., 2022). In addition, normalizing flows preserve the dimensionality of the inputs, requiring a latent space of the same dimension as the data space.

Due to the manifold hypothesis (Bengio et al., 2013), which suggests that realistic data lies on a low-dimensional manifold embedded into a high-dimensional data space, it is more efficient to only model distributions on a low-dimensional manifold and regard deviations from the manifold as uninformative noise. Prior works such as Caterini et al. (2021); Brehmer & Cranmer (2020) have restricted normalizing flows to low-dimensional manifolds via specially-constructed bottleneck architectures (known as "invertible autoencoders" (Teng & Choromanska, 2019) or "injective flows" (Kothari et al., 2021)) where encoder and decoder share parameters. These injective flows are optimized by some version of maximum likelihood training. This is not an ideal design decision, as the restrictive architectures used in such models were originally designed for tractable change of variables calculations in normalizing flows, but such calculations are not possible in the presence of a bottleneck (Brehmer & Cranmer, 2020). As a result, we propose to drop the restrictive constructions (such as coupling blocks), instead use an unconstrained encoder and decoder, and introduce a new technique to get around calculating the change of variables. This greatly simplifies the design of the model and makes it more expressive.

---

[*]Equal contribution.

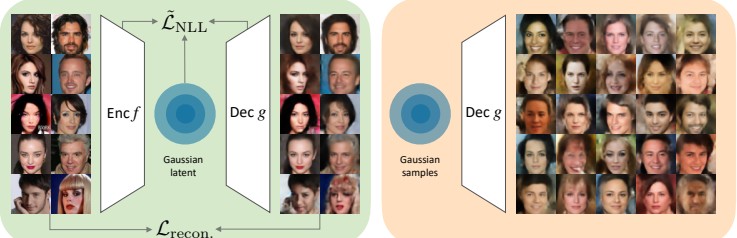

Figure 1: **Free-form injective flow (FIF) training and inference.** *(Left)* We combine a reconstruction loss $\mathcal{L}_{\text{recon.}}$ with a novel maximum likelihood loss $\tilde{\mathcal{L}}_{\text{NLL}}$ to obtain an injective flow without architectural constraints. *(Right)* We generate novel samples by decoding standard normal latent samples with our best-performing models on CelebA and MNIST. The reconstructions shown are on CelebA validation data, the samples are uncurated samples from our models.

We build on the unbiased maximum likelihood estimator used by rectangular flows (Caterini et al., 2021) to approximate the gradient of the change of variables term. We simplify the estimator considerably by replacing iterative conjugate gradient with an efficient single-step estimator. This is fast: a batch can be processed in about twice the time (or less) as an autoencoder trained on reconstruction loss only. In addition, we make a novel observation about injective flows: naively training with maximum likelihood is ill-defined due to the possibility of diverging curvature in the decoding function. To fix this problem, we propose a modification to our maximum likelihood estimator which counteracts the possibility of diverging curvature. We call our model the *free-form injective flow* (FIF).

To summarize, we make the following contributions:

- We introduce an efficient maximum likelihood estimator for free-form injective flows and use it to train an unconstrained injective flow for the first time (section 4.1).
- We identify pathological behavior in the naive application of maximum likelihood training in the presence of a bottleneck, and offer a solution to avoid this behavior while maintaining computational efficiency (section 4.2).
- We outperform previous injective flows and demonstrate competitive performance to generative autoencoders on toy, tabular and image data (section 5). We provide code to implement our model and reproduce our results at `https://github.com/vislearn/FFF`.

## 2 RELATED WORK

Injective flows jointly learn a manifold and maximize likelihood on that manifold. The latter requires estimating the Jacobian determinant of the transformation to calculate the change of variables. Efficient computation of this determinant traditionally imposed two major restrictions on normalizing flow architectures: Firstly, the latent space has to match in dimension with the data space, ruling out bottleneck architectures. Secondly, normalizing flows are restricted to certain functional forms, such as coupling and autoregressive blocks. Below we outline the existing approaches to overcome these problems and how our solution compares.

**Lower-dimensional latent spaces** One set of methods attempts to use full-dimensional normalizing flows, with some additional regularization or architectural constraints such that a subspace of the latent space corresponds to the manifold. One strategy adds noise to the data to make it a full-dimensional distribution then denoises to the manifold (Horvat & Pfister, 2021; Loaiza-Ganem et al., 2022). Another restricts the non-manifold latent dimensions to have small variance (Beitler et al., 2021; Silvestri et al., 2023; Zhang et al., 2023).

Other methods sidestep the problem by making training into a two-step procedure. First, an autoencoder is trained on reconstruction loss, then a normalizing flow is trained to learn the resulting latent distribution. In this line of work, Brehmer & Cranmer (2020) and Kothari et al. (2021) use

an injective flow, while Böhm & Seljak (2022) use unconstrained networks as autoencoder. Ghosh et al. (2020) additionally regularize the decoder.

Conformal embedding flows (Ross & Cresswell, 2021) ensure the decomposition of the determinant into the contribution from each block by further restricting the architecture to exclusively conformal transformations. Cramer et al. (2022) uses an isometric autoencoder such that the change of variables is trivial. However, the resulting transformations are quite restrictive and cannot represent arbitrary manifolds.

The most similar work to ours is the rectangular flow (Caterini et al., 2021) which estimates the gradient of the log-determinant via an iterative, unbiased estimator. The resulting method is quite slow to train, and uses injective flows, which are restrictive.

**Unconstrained normalizing flow architectures**    Several works attempt to reduce the constraints imposed by typical normalizing flow architectures, allowing the use of free-form networks. However, all of these methods only apply to full-dimensional architectures. FFJORD (Grathwohl et al., 2019) is a type of continuous normalizing flow (Chen et al., 2018b) which estimates the change of variables stochastically. Residual flows (Behrmann et al., 2019; Chen et al., 2019) make residual networks invertible, but require expensive iterative estimators to train via maximum likelihood. Self-normalizing flows (Keller et al., 2021) and relative gradient optimization (Gresele et al., 2020) estimate maximum likelihood gradients for the matrices used in neural networks, but restrict the architecture to use exclusively square weight matrices without skip connections. In a parallel work, we present the extension of the present paper to full-dimensional normalizing flows based on free-form neural networks, called free-form flows (Draxler et al., 2024).

**Approximating maximum likelihood**    Many methods optimize some bound on the full-dimensional maximum likelihood, notably the variational autoencoder (Kingma & Welling, 2014) and its variants. Cunningham et al. (2020) also optimizes a variational lower bound to the likelihood. Other methods fit into the injective flow framework by jointly optimizing a reconstruction loss and some approximation to maximum likelihood on the manifold: Kumar et al. (2020); Zhang et al. (2020) approximate the log-determinant of the Jacobian by its Frobenius norm. The entropic AE (Ghose et al., 2020) maximizes the entropy of the latent distribution by a nearest-neighbor estimator while constraining its variance, resulting in a Gaussian latent space. In addition, there are other ways to regularize the latent space of an autoencoder which are not based on maximum likelihood, e.g. adversarial methods (Makhzani et al., 2015).

In contrast to the above, our approach jointly learns the manifold and maximizes the likelihood on it with an unconstrained architecture, which easily accommodates a lower-dimensional latent space.

## 3    BACKGROUND

**Notation**    Let $f : \mathbb{R}^D \to \mathbb{R}^d$ be an encoder which compresses data to a latent space and a decoder $g : \mathbb{R}^d \to \mathbb{R}^D$ which decompresses the latent representation. A full-dimensional model has $d = D$ while a bottleneck model has $d < D$. If $f \circ g : \mathbb{R}^d \to \mathbb{R}^d$ is the identity, then we call $f$ and $g$ *consistent*. For example, the forward and inverse function of a normalizing flow are consistent as $f^{-1} = g$.

**Injective flows**    Injective flows (Brehmer & Cranmer, 2020), also called invertible autoencoders (Teng & Choromanska, 2019), adapt invertible normalizing flow architectures to a bottleneck setting. They parameterize $f$ and $g$ as the composition of two invertible functions, $w$ defined in $\mathbb{R}^D$ and $h$ defined in $\mathbb{R}^d$, with a slicing/padding operation in between:

$$f = h^{-1} \circ \mathtt{slice} \circ w^{-1} \quad \text{and} \quad g = w \circ \mathtt{pad} \circ h, \tag{1}$$

where $\mathtt{slice}(x)$ selects the first $d$ elements of $x$ and $\mathtt{pad}(z)$ concatenates $D - d$ zeros to the end of $z$. Since $\mathtt{slice}$ and $\mathtt{pad}$ are consistent, so too are $f$ and $g$.

Injective flows typically minimize a reconstruction loss (to learn a manifold which spans the data) alongside maximizing the likelihood of the data on that manifold, performing joint manifold and maximum likelihood training (alternatively, they are trained via two-step training, see section 2)

**Change of variables across dimensions** The maximum likelihood objective resulting from the change of variables theorem, used to train normalizing flow models, is only well-defined when mapping between spaces of equal dimension. A result from differential geometry (Krantz & Parks, 2008) allows us to generalize the change of variables theorem to non-equal-dimension transformations through the formula:

$$p_X(x) = p_Z(f(x)) \left( \det \left[ g'(f(x))^\top g'(f(x)) \right] \right)^{-\frac{1}{2}}, \tag{2}$$

where $f$ and $g$ are consistent and primes denote derivatives: $g'(f(x))$ is the Jacobian matrix of $g$ evaluated at $f(x)$. Note that, since $p_X$ is derived as the pushforward of the latent distribution $p_Z$ by $g$, the formula is valid only for $x$ which lie on the decoder manifold (see appendix A for more details). Unlike in the full-dimensional case, there is no architecture known which can represent arbitrary manifolds and at the same time allows efficient exact computation of eq. (2) (see section 2).

**Rectangular flows** Minimizing the negative logarithm of eq. (2) and adding a Lagrange multiplier to restrict the distance of data points from the decoder manifold results in the following per-sample loss term:

$$\mathcal{L}_{\text{RF}}(x) = -\log p_Z(z) + \frac{1}{2} \log \det \left[ g'(z)^\top g'(z) \right] + \beta \|\hat{x} - x\|^2, \tag{3}$$

where $z = f(x)$, $\hat{x} = g(z)$, and $\beta$ is a hyperparameter.

The log-determinant term is the difficult one to optimize. Fortunately, its gradient with respect to the parameters $\theta$ of the decoder can be estimated tractably by the following construction (Caterini et al., 2021). Note that $g = g_\theta$ but the $\theta$ subscript is dropped to avoid clutter. The relevant quantity is (with $J = g'(z)$):

$$\frac{\partial}{\partial \theta_j} \frac{1}{2} \log \det(J^\top J) = \frac{1}{2} \operatorname{tr} \left( (J^\top J)^{-1} \frac{\partial}{\partial \theta_j} (J^\top J) \right). \tag{4}$$

The trace can be estimated (Hutchinson, 1989; Girard, 1989) by:

$$\frac{\partial}{\partial \theta_j} \frac{1}{2} \log \det(J^\top J) \approx \frac{1}{2K} \sum_{k=1}^{K} \underbrace{\epsilon_k^\top (J^\top J)^{-1}}_{(**)} \frac{\partial}{\partial \theta_j} \underbrace{(J^\top J) \epsilon_k}_{(*)}, \tag{5}$$

where the $\epsilon_k$ are $K$ samples from a distribution where $E[\epsilon \epsilon^\top] = \mathbb{I}$, typically either Rademacher or standard normal. Now, we take steps to compute the above without constructing the entire network Jacobian $J$. First, note that $(*)$ can be written as a Jacobian-vector product $v_1 = Jv$ and a vector-Jacobian product $J^T v_1 = (v_1^\top J)^\top$, each readily available via automatic differentiation. For $(**)$, Caterini et al. (2021) propose to employ the iterative conjugate gradient method: Write $\epsilon_k^\top (J^\top J)^{-1} = \text{CG}(J^\top J; \epsilon_k)^\top$ where $\text{CG}(A; b)$ denotes the conjugate gradient solution to $Ax = b$. Conjugate gradient is a clever choice here, since it again only requires computing terms of the form $J^\top Jv$ using autodiff. The parameter derivative can be made to act only on the rightmost Jacobian terms by applying the `stop_gradient` operation to the output of the conjugate gradient method, which returns its input, but has zero gradient. The final surrogate for the log-determinant term is therefore:

$$\frac{1}{2K} \sum_{k=1}^{K} \texttt{stop\_gradient} \left( \text{CG} \left( J^\top J; \epsilon_k \right)^\top \right) J^\top J \epsilon_k, \tag{6}$$

which replaces the log-determinant term in the loss. This computation yields the same gradient as the original loss in eq. (3), even though it has a different value. In the following, we present a significantly improved version of this estimator based on the insight that the encoder is an (approximate) inverse of the decoder.

## 4 FREE-FORM INJECTIVE FLOW (FIF)

Our modification to rectangular flows is threefold: first, we use an unconstrained autoencoder architecture (no restrictively parameterized invertible functions); second, we introduce a more computationally efficient surrogate estimator; third, we modify the surrogate to avoid pathological behavior

related to manifolds with high curvature. Our per-sample loss function is:

$$\mathcal{L}(x) = -\log p_Z(z) - \frac{1}{K}\sum_{k=1}^{K}\epsilon_k^\top f'(x)\,\texttt{stop\_gradient}\,(g'(z)\epsilon_k) + \beta\|\hat{x} - x\|^2, \quad (7)$$

with $z = f(x)$. Note the negative sign before the surrogate term, which comes from sending the log-determinant gradient to the encoder rather than the decoder. We will derive and motivate this formulation of the loss in sections 4.1 and 4.2.

## 4.1 SIMPLIFYING THE SURROGATE

We considerably simplify the optimization of rectangular flows by a new surrogate for the log-determinant term in eq. (2), which uses the Jacobian of the encoder as an approximation for the inverse Jacobian of the decoder. This allows the surrogate to be computed in a single pass, avoiding costly conjugate gradient iterations.

We do this by expanding the derivative in eq. (4):

$$\frac{1}{2}\,\mathrm{tr}\left((J^\top J)^{-1}\frac{\partial}{\partial\theta_j}(J^\top J)\right) = \mathrm{tr}\left(J^\dagger\frac{\partial}{\partial\theta_j}J\right), \quad (8)$$

where $J^\dagger = (J^\top J)^{-1}J^\top$ is the Moore-Penrose inverse of $J$. The full derivation is in appendix B. To see the advantage of this formulation consider that for encoder $f$ and decoder $g$ optimal with respect to the reconstruction loss, $f'(\hat{x}) = g'(f(x))^\dagger$ (see appendix B.1). Using the $\texttt{stop\_gradient}$ operation, this leads to the following surrogate loss term:

$$\frac{1}{K}\sum_{k=1}^{K}\texttt{stop\_gradient}\left(\epsilon_k^\top f'(\hat{x})\right)g'(z)\epsilon_k, \quad (9)$$

or equivalently, using the encoder Jacobian in place of the decoder Jacobian (see appendix B):

$$-\frac{1}{K}\sum_{k=1}^{K}\epsilon_k^\top f'(\hat{x})\texttt{stop\_gradient}\left(g'(z)\epsilon_k\right). \quad (10)$$

Each term of the sum can be computed from just two vector-Jacobian/Jacobian-vector products obtained from automatic differentiation. This is a significant improvement on the iterative conjugate gradient method needed in the original formulation of rectangular flows which requires up to $2(d+1)$ such products to ensure convergence (Caterini et al., 2021). Compared to reconstruction loss only, we measure $\sim 1.5\times$ to $2\times$ the wall clock time, independent of the latent dimension. Note that the surrogate is only accurate if $f$ and $g$ are (at least approximately) optimal with respect to the reconstruction loss. We observe stable training in practice, validating this assumption.

## 4.2 PROBLEMS WITH MAXIMUM LIKELIHOOD IN THE PRESENCE OF A BOTTLENECK

Rectangular flows are trained with a combination of a reconstruction and a likelihood term. We might ask what happens if we only train with the likelihood term, making an analogy to normalizing flows. In this case our loss would be:

$$\mathcal{L}_{\mathrm{NLL}}(x) = -\log p_Z(z) + \frac{1}{2}\log\det\left[g'(z)^\top g'(z)\right]. \quad (11)$$

Unfortunately, optimizing this loss can lead us to learn a degenerate decoder manifold, an issue raised in Brehmer & Cranmer (2020). Here we expand on their argument. First consider that if $f$ and $g$ are consistent, then $f(\hat{x}) = f(g(f(x))) = f(x)$ and the per-sample loss is invariant to projections: $\mathcal{L}_{\mathrm{NLL}}(\hat{x}) = \mathcal{L}_{\mathrm{NLL}}(x)$, since $\mathcal{L}_{\mathrm{NLL}}(x)$ is a function only of $f(x)$. This means that we can write our loss as:

$$\mathcal{L}_{\mathrm{NLL}} = E_{p_{\mathrm{data}}(x)}[\mathcal{L}_{\mathrm{NLL}}(x)] = E_{\hat{p}_{\mathrm{data}}(\hat{x})}[\mathcal{L}_{\mathrm{NLL}}(\hat{x})], \quad (12)$$

where $\hat{p}_{\mathrm{data}}(\hat{x})$ is the probability density of the projection of the training data onto the decoder manifold. Now consider that the negative log-likelihood loss is one part of a KL divergence, and KL divergences are always non-negative:

$$\mathrm{KL}(\hat{p}_{\mathrm{data}}(\hat{x})\|p_\theta(\hat{x})) = -H(\hat{p}_{\mathrm{data}}(\hat{x})) - E_{\hat{p}_{\mathrm{data}}(\hat{x})}[\log p_\theta(\hat{x})] \geq 0. \quad (13)$$

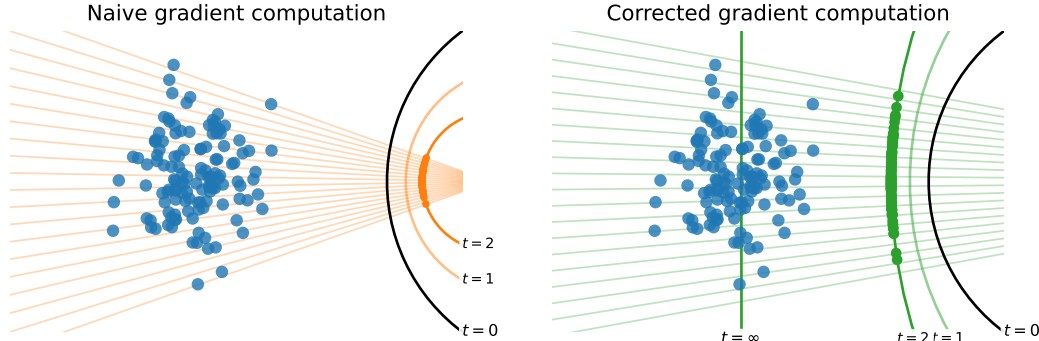

Figure 2: Naive training of autoencoders with negative log-likelihood (NLL, see section 4.2) leads to pathological solutions *(left)*. Starting with the initialization ($t = 0$, black), gradient steps increase the curvature of the learnt manifold ($t = 1, 2$, orange). This reduces NLL because the entropy of the projected data is reduced, by moving the points closer to one another. This effect is stronger than the reconstruction loss. We fix this problem by evaluating the volume change off-manifold *(right)*. This moves the manifold closer to the data and reduces the curvature ($t = 1, 2$, green), until it eventually centers the manifold on the data with zero curvature ($t = \infty$, green). Light lines show the set of points which map to the same latent point. Data is projected onto the $t = 2$ manifold.

As a result, the loss is lower bounded by the entropy of the data projected onto the manifold:

$$\mathcal{L}_{\text{NLL}} = -E_{\hat{p}_{\text{data}}(\hat{x})}[\log p_\theta(\hat{x})] \geq H(\hat{p}_{\text{data}}(\hat{x})). \tag{14}$$

Unlike in standard normalizing flow optimization, where the right hand side would be fixed, here the entropy depends on the projection learned by the model. Thus, the model could modify the projection such that entropy is as low as possible. We break this pathology down into two cases:

1. *A model manifold which does not align with the data manifold but instead intersects it.* For example, Brehmer & Cranmer (2020) discuss a case where a linear model learns to project a data distribution to a single point on the manifold, thus reducing its entropy to $-\infty$, the lowest possible value. To the best of our knowledge, this can be fixed by adding noise and a reconstruction loss with sufficiently high weight. In appendix C we prove as much for linear models and characterize the solutions, which are the same as PCA if $\beta \geq 1/2\sigma^2$ where $\sigma^2$ is the smallest eigenvalue of the data covariance matrix.

2. *A model manifold which concentrates the data by use of high curvature,* see fig. 2 *(left)*. This newly identified pathological case only occurs in nonlinear models, hence Brehmer & Cranmer (2020) did not notice this effect in their linear example. Importantly, this is **not** fixed by adding a reconstruction loss.

Most existing injective flows avoid this by a two-stage training, which first learns a projection and then the distribution of the projected data in the latent space. To enable jointly learning a manifold and a maximum-likelihood density on it, we need to find a fix for the pathology.

**Towards a well-behaved loss**    The term which leads to pathological behavior in the likelihood loss is the log-determinant. When using the change of variables with $f'$ evaluated at $\hat{x}$, all that matters is the change of volume from the projected data to the latent space, so the model can decrease the loss by choosing a manifold which concentrates the projected data more tightly (the more possibility it has to expand the data, the lower the loss will be). We can counteract this effect by introducing a factor inversely proportional to the concentration. This can be achieved by the fairly simple modification of evaluating $f'$ in our estimator at $x$ rather than $\hat{x}$. Namely, we modify eq. (10) to estimate the gradient of the log-determinant term by:

$$-\frac{1}{K}\sum_{k=1}^{K} \epsilon_k^\top f'(x)\,\texttt{stop\_gradient}\left(g'(z)\epsilon_k\right). \tag{15}$$

See appendix B.2 for a detailed explanation.

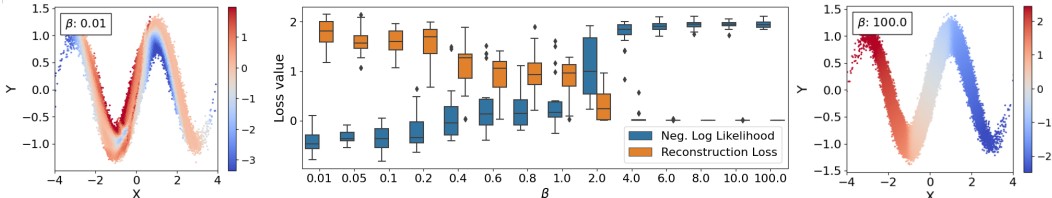

Figure 3: Learning a noisy 2-D sinusoid with a 1-D latent space for different reconstruction weights $\beta$. Color codes denote the value of the latent variable at each location. When the reconstruction term has low weight (*left*), the autoencoder learns to throw away information about the position along the sinusoid and only retains the orthogonal noise. Only sufficiently high weights (*right*) result in the desired solution, where the decoder spans the sinusoid manifold. The middle plot shows the tradeoff between reconstruction error and NLL as we transition between these regimes (box plots indicate variability across runs).

In this way, we discourage pathological solutions involving high curvature. In fig. 2 (*right*) we can see the effect of the modified estimator: the manifold now moves towards the data since the optimization is not dominated by diverging curvature. We note that the modified estimator is also computationally cheaper, since the vector-Jacobian product $\epsilon_k^\top f'(x)$ can reuse the computational graph generated when computing $z$.

Along with the results of section 4.1, this leads to the following loss (same as eq. (7)):

$$\mathcal{L}(x) = \tilde{\mathcal{L}}_{\text{NLL}}(x) + \beta \mathcal{L}_{\text{recon.}}(x) \tag{16}$$

$$= -\log p_Z(z) - \frac{1}{K} \sum_{k=1}^{K} \epsilon_k^\top f'(x) \, \texttt{stop\_gradient}\left(g'(z)\epsilon_k\right) + \beta \|\hat{x} - x\|^2. \tag{17}$$

**Phase transition** Figure 3 shows that when using this loss, if $\beta$ is large enough, the dominant manifold direction is identified. In appendix E.1.2, we show a similar experiment on MNIST.

## 5 EXPERIMENTS

In this section, we test the empirical performance of the proposed model. First, we show that our model is much faster than rectangular flows on tabular data. Second, we show that it outperforms previous SOTA injective flows on generating images. Finally, we compare against other generative autoencoders on the Pythae image generation benchmark Chadebec et al. (2022), achieving the best FID score in some categories.

**Implementation details** In implementing the trace estimator, we have to make a number of choices. Briefly, i) we chose to formulate the log-determinant gradient in terms of the encoder rather than decoder as it was more stable in practice, ii) we performed traces in the order $f'(x)g'(z)$ as this reduces variance (both orderings are valid due to the cyclic property of the trace but since $f'(x)g'(z)$ is a $d \times d$ matrix whereas $g'(z)f'(x)$ is $D \times D$, the former is typically easier to estimate), iii) we used a mixture of forward- and backward-mode automatic differentiation as this was compatible with our estimator, and iv) we used orthogonalized Gaussian noise in the trace estimator, to reduce variance. Full justification for these choices is given in appendix D.

### 5.1 TABULAR DATA

We evaluate our method on four of the tabular datasets used by Papamakarios et al. (2017), using the same data splits, and make a comparison to the published rectangular flow results (Caterini et al., 2021), see table 1. We adopt the "FID-like metric" from that work, which computes the Wasserstein-2 distance between the Gaussian distributions with equal mean and covariance as the test data and the data generated by the model. This is a measure of the difference of the means and covariance matrices of the generated and test datasets. We outperform rectangular flows on all datasets except GAS. In addition, we see a speedup in training time of between 1.5 and 6 times between FIF and

Table 1: Free-form injective flows (FIF) are significantly faster than rectangular flows (RF) with superior performance in FID-like metric on 3 out of 4 tabular datasets (Papamakarios et al., 2017). Both methods use $K = 1$. The results for RF are taken directly from (Caterini et al., 2021).

| Method | POWER | GAS | HEPMASS | MINIBOONE |
|---|---|---|---|---|
| RF (Caterini et al., 2021) | $0.083 \pm 0.015$ | $\mathbf{0.110 \pm 0.021}$ | $0.779 \pm 0.191$ | $1.001 \pm 0.051$ |
| FIF *(ours)* | $\mathbf{0.041 \pm 0.007}$ | $0.281 \pm 0.031$ | $\mathbf{0.541 \pm 0.034}$ | $\mathbf{0.598 \pm 0.024}$ |
| Training Time Speedup | $\mathbf{3.9} \times$ | $\mathbf{2.2} \times$ | $\mathbf{6.1} \times$ | $\mathbf{1.5} \times$ |

a rerun of rectangular flows (using the published code) on the same hardware. Full experimental details are in appendix E.2.

We also show an ablation study in table 6 and table 7 in the appendix disentangling the effect of the three individual components of our method: The surrogate of eq. (10), the fix for high curvature solutions in eq. (15) and the use of free-form architectures. Comparing to table 1, we find that the fix to estimate the negative log-likelihood with an off-manifold encoder Jacobian is crucial for good performance of free-from architectures, as the on-manifold variant diverges (see section 4.2). This is not the case for the injective architecture based on coupling flows, indicating a stabilizing regularization via the architecture.

## 5.2 COMPARISON TO INJECTIVE FLOWS

We compare FIF against previous injective flows on CelebA images (Liu et al., 2015) in table 2. Our models significantly improve the quality of the generated images in terms of the Fréchet inception distance (FID) Heusel et al. (2017) and Inception Score (IS) Salimans et al. (2016). The former compares generated samples to a set of reference samples, by computing the Wasserstein-2 distance between two Gaussian distributions fit to some embedding of the respective sets of samples. The later measures diversity by the entropy of the distribution of class labels in the generated samples, where the class labels are provided by some pre-trained classifier. Samples from this model are depicted in fig. 1.

For a fair comparison, we train each model on the same hardware for equal wall clock time with the code provided by the authors. The architectures of previous works were dominated by the need that most layers are invertible and have a tractable Jacobian determinant. Our loss in eq. (7) does not impose these constraints on the architecture, and we can use an off-the-shelf convolutional auto-encoder with additional fully-connected layers in the latent space. Details can be found in appendix E.3.

## 5.3 COMPARISON TO GENERATIVE AUTOENCODERS

As free-form injective flows (FIF) do not require any specific architecture, we expand our comparison to the much broader range of *generative autoencoders*. This is a general class of bottleneck architectures that encode the training data to a standard normal distribution, so that the decoder can be used as a generator after training.

Recently, Chadebec et al. (2022) proposed the Pythae benchmark for comparing generative autoencoders on image generation. They evaluate different training methods using two different architec-

Table 2: **Comparison of injective flows on CelebA** under equal computational budget. Free-form Injective Flows (FIF) outperform previous work significantly in terms of FID.

| Model | # parameters | $\mathcal{N}$ sampler | | GMM sampler | |
|---|---|---|---|---|---|
| | | FID $\downarrow$ | IS $\uparrow$ | FID $\downarrow$ | IS $\uparrow$ |
| DNF (Horvat & Pfister, 2021) | 39.4M | $55.6 \pm 0.59$ | 1.9 | $52.7 \pm 0.33$ | 2.0 |
| Trumpet (Kothari et al., 2021) | 19.1M | $56.2 \pm 1.39$ | 1.8 | $47.7 \pm 2.24$ | 1.9 |
| FIF *(ours)* | 34.3M | $\mathbf{47.3 \pm 1.39}$ | 1.7 | $\mathbf{37.4 \pm 1.35}$ | 2.0 |

Table 3: **Pythae benchmark results on CelebA**, following Chadebec et al. (2022). We train their architectures (ConvNet and ResNet) with our new training objective, achieving SOTA FID on ResNet. We draw latent samples from standard normal "$\mathcal{N}$" or a GMM fit using training data "GMM". Models with multiple variants (indicated in brackets) have been merged to indicate only the best result across variants. We mark the best FID in each column in bold and underline the second best.

| Model | ConvNet + $\mathcal{N}$ | | ResNet + $\mathcal{N}$ | | ConvNet + GMM | | ResNet + GMM | |
| --- | --- | --- | --- | --- | --- | --- | --- | --- |
| | FID $\downarrow$ | IS $\uparrow$ | FID | IS | FID | IS | FID | IS |
| VAE (Kingma & Welling, 2014) | 54.8 | 1.9 | 66.6 | 1.6 | 52.4 | 1.9 | 63.0 | 1.7 |
| IWAE (Burda et al., 2015) | 55.7 | 1.9 | 67.6 | 1.6 | 52.7 | 1.9 | 64.1 | 1.7 |
| VAE-lin NF (Rezende & Mohamed, 2015) | 56.5 | 1.9 | 67.1 | 1.6 | 53.3 | 1.9 | 62.8 | 1.7 |
| VAE-IAF (Kingma et al., 2016) | 55.4 | 1.9 | 66.2 | 1.6 | 53.6 | 1.9 | 62.7 | 1.7 |
| $\beta$-(TC) VAE (Higgins et al., 2017; Chen et al., 2018a) | 55.7 | 1.8 | 65.9 | 1.6 | 51.7 | 1.9 | 59.3 | 1.7 |
| FactorVAE (Kim & Mnih, 2018) | **53.8** | 1.9 | 66.4 | 1.7 | 52.4 | 2.0 | 63.3 | 1.7 |
| InfoVAE - (RBF/IMQ) (Zhao et al., 2017) | 55.5 | 1.9 | 66.4 | 1.6 | 52.7 | 1.9 | 62.3 | 1.7 |
| AAE (Makhzani et al., 2015) | 59.9 | 1.8 | 64.8 | 1.7 | 53.9 | 2.0 | 58.7 | 1.8 |
| MSSSIM-VAE (Snell et al., 2017) | 124.3 | 1.3 | 119.0 | 1.3 | 124.3 | 1.3 | 119.2 | 1.3 |
| Vanilla AE | 327.7 | 1.0 | 275.0 | 2.9 | 55.4 | 2.0 | 57.4 | 1.8 |
| WAE - (RBF/IMQ) (Tolstikhin et al., 2018) | 64.6 | 1.7 | 67.1 | 1.6 | 51.7 | 2.0 | 57.7 | 1.8 |
| VQVAE (van den Oord et al., 2017) | 306.9 | 1.0 | 140.3 | 2.2 | 51.6 | 2.0 | 57.9 | 1.8 |
| RAE - (L2/GP) Ghosh et al. (2020) | 86.1 | 2.8 | 168.7 | 3.1 | 52.5 | 1.9 | 58.3 | 1.8 |
| FIF *(ours)* | 56.9 | 2.1 | **62.3** | 1.7 | **47.3** | 1.9 | **55.0** | 1.8 |

tures on MNIST (LeCun et al., 2010) (data $D = 784$, latent $d = 16$), CIFAR10 (Krizhevsky, 2009) ($D = 3072, d = 256$), and CelebA (Liu et al., 2015) ($D = 12288, d = 64$). All models are trained with the same limited computational budget. The goal of the benchmark is to provide a fair comparison of different models, not to achieve SOTA image generation results, as this would require significantly more compute.

As shown in table 3, our model performs strongly on the benchmark, achieving SOTA on CelebA in Fréchet Inception Distance (FID) (Heusel et al., 2017) on the ResNet architecture with latent codes sampled from a standard normal, and on both architectures when sampling from a Gaussian Mixture Model fit using training data. At the same time, the Inception Scores (IS) (Salimans et al., 2016) are high, indicating a high diversity. On the one combination where our model does not outperform the competitors, FIF still achieves a comparable FID and high Inception Score. FIF also performs strongly on the other datasets, see appendix E.4.

For each method in the benchmark, ten hyperparameter configurations are trained and the best model according to FID is reported. For our method, we choose to vary $\beta = 5, 10, 15, 20, 25$ and the number of Hutchinson samples $K = 1, 2$. We find the performance to be robust against these choices, and give all details on the training procedure in appendix E.4.

## 6  CONCLUSION

This paper offers a computationally efficient solution to jointly learning a manifold and a distribution on it, which we call the free-form injective flow (FIF). We i) significantly improve an existing estimator for the gradient of the change of variables across dimensions, ii) note that it can be applied to unconstrained architectures, iii) analyze problems with joint manifold and maximum-likelihood training and offer a solution, and iv) implement and test our model on toy, tabular and image datasets. We find that the model is practical and scalable, outperforming comparable injective flows, and showing similar or better performance to other autoencoder generative models on the Pythae benchmark.

Several theoretical and practical questions remain for future work: We identified a previously overlooked problem with jointly learning a manifold and maximum likelihood. We propose a fix in section 4.2 that provides high-quality models, but further investigation is needed for a thorough understanding. Fitting a GMM to the latent space after training improves performance on image data, suggesting that our latent distributions are not perfectly Gaussian. We generally find that architectures with more fully-connected layers in the latent space have a more Gaussian latent distribution, suggesting that larger models suffer less from this problem. We leave potential theoretical or practical improvements to future work.

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

# A    CHANGE OF VARIABLES FORMULA ACROSS DIMENSIONS

The change of variables formula describes how probability densities change as they are mapped through an injective "pushforward" function $g$. It is instructive to derive this formula when $g$ is an invertible function. Let $p_Z$ be a base density and $p_X$ the pushforward density obtained by mapping samples from $p_Z$ through $g$. Then we can write

$$p_X(x) = \int p(x \mid z)p_Z(z)\mathrm{d}z \tag{18}$$

$$= \int \delta(x - g(z))p_Z(z)\mathrm{d}z \tag{19}$$

$$= \int \delta(x - \hat{x})p_Z(f(\hat{x})) \left|\det(g'(f(\hat{x})))\right|^{-1} \mathrm{d}\hat{x} \tag{20}$$

$$= p_Z(f(x)) \left|\det(g'(f(x)))\right|^{-1} \tag{21}$$

using the change of variables $\hat{x} = g(z)$, meaning that $z = f(\hat{x})$ and $|\det(g'(z))|\mathrm{d}z = \mathrm{d}\hat{x}$ with $f$ the inverse of $g$.

Now suppose that $g$ maps from $\mathbb{R}^d$ to $\mathbb{R}^D$ with $d < D$. We can generalize the change of variables $\hat{x} = g(z)$ using $z = f(\hat{x})$ and $\det(g'(z)^\top g'(z))^{\frac{1}{2}}\mathrm{d}z = \mathrm{d}\hat{x}$ where $f$ and $g$ are consistent ($f \circ g$ is the identity) (see chapter 5 of Krantz & Parks (2008)). This gives us

$$p_X(x) = \int \delta(x - g(z))p_Z(z)\mathrm{d}z \tag{22}$$

$$= \int \delta(x - \hat{x})p_Z(f(\hat{x})) \det(g'(f(\hat{x}))^\top g'(f(\hat{x})))^{-\frac{1}{2}} \mathrm{d}\hat{x} \tag{23}$$

This expression defines a probability density in the full ambient space $\mathbb{R}^D$ (albeit a degenerate distribution) but we cannot easily remove the integral. However, we can convert it into an expression resembling the full-dimensional case, but defined only on the image of $g$:

$$p_X(x) = p_Z(f(x)) \det(g'(f(x))^\top g'(f(x)))^{-\frac{1}{2}} \tag{24}$$

Note that this expression only integrates to 1 if we restrict integration to the image of $g$. As such, it should only be regarded as defining a probability distribution on this manifold, not in the ambient space $\mathbb{R}^D$.

# B    DERIVATION OF GRADIENT ESTIMATOR

We expand the derivative in eq. (4):

$$\frac{\partial}{\partial\theta_j} \frac{1}{2} \log\det J^\top J = \frac{1}{2} \operatorname{tr}\left((J^\top J)^{-1} \frac{\partial}{\partial\theta_j}(J^\top J)\right) \tag{25}$$

$$= \frac{1}{2} \operatorname{tr}\left((J^\top J)^{-1} \left(\frac{\partial}{\partial\theta_j} J^\top\right) J\right) + \frac{1}{2} \operatorname{tr}\left((J^\top J)^{-1} J^\top \left(\frac{\partial}{\partial\theta_j} J\right)\right) \tag{26}$$

$$= \frac{1}{2} \operatorname{tr}\left((J(J^\top J)^{-1})^\top \frac{\partial}{\partial\theta_j} J\right) + \frac{1}{2} \operatorname{tr}\left((J^\top J)^{-1} J^\top \frac{\partial}{\partial\theta_j} J\right) \tag{27}$$

$$= \operatorname{tr}\left((J^\top J)^{-1} J^\top \frac{\partial}{\partial\theta_j} J\right) \tag{28}$$

$$= \operatorname{tr}\left(J^\dagger \frac{\partial}{\partial\theta_j} J\right) \tag{29}$$

where we used the cyclic property of the trace and that $\operatorname{tr}(AB) = \operatorname{tr}(A^\top B^\top)$. $J^\dagger = (J^\top J)^{-1} J^\top$ is the Moore-Penrose inverse of $J$.

Now we will do an equivalent derivation for the encoder. Observe that, since

$$(J^\dagger J^{\dagger\top})^{-1} = ((J^\top J)^{-1} J^\top J (J^\top J)^{-1})^{-1} = J^\top J \tag{30}$$

we can rewrite the log-determinant term using the encoder Jacobian:

$$\frac{1}{2} \log \det(J^\top J) = -\frac{1}{2} \log \det(J^\dagger J^{\dagger\top}) \tag{31}$$

Note the negative sign on the right-hand side.

The derivation for the derivative is very similar to that for the decoder, where we now take a derivative with respect to encoder parameters $\phi$:

$$\frac{\partial}{\partial \phi_j} \frac{1}{2} \log \det J^\dagger J^{\dagger\top} = \frac{1}{2} \operatorname{tr}\left( (J^\dagger J^{\dagger\top})^{-1} \frac{\partial}{\partial \phi_j} (J^\dagger J^{\dagger\top}) \right) \tag{32}$$

$$= \frac{1}{2} \operatorname{tr}\left( (J^\dagger J^{\dagger\top})^{-1} \left( \frac{\partial}{\partial \phi_j} J^\dagger \right) J^{\dagger\top} \right) + \frac{1}{2} \operatorname{tr}\left( (J^\dagger J^{\dagger\top})^{-1} J^\dagger \left( \frac{\partial}{\partial \phi_j} J^{\dagger\top} \right) \right) \tag{33}$$

$$= \frac{1}{2} \operatorname{tr}\left( J^{\dagger\top} (J^\dagger J^{\dagger\top})^{-1} \frac{\partial}{\partial \phi_j} J^\dagger \right) + \frac{1}{2} \operatorname{tr}\left( \left( (J^\dagger J^{\dagger\top})^{-1} J^\dagger \right)^\top \frac{\partial}{\partial \phi_j} J^\dagger \right) \tag{34}$$

$$= \operatorname{tr}\left( J^{\dagger\top} (J^\dagger J^{\dagger\top})^{-1} \frac{\partial}{\partial \phi_j} J^\dagger \right) \tag{35}$$

$$= \operatorname{tr}\left( J \frac{\partial}{\partial \phi_j} J^\dagger \right) \tag{36}$$

where we used the cyclic property of the trace, that $\operatorname{tr}(AB) = \operatorname{tr}(A^\top B^\top)$ and that $J = \left( J^\dagger \right)^\dagger = J^{\dagger\top}(J^\dagger J^{\dagger\top})^{-1}$.

Recall eq. (9) in section 4.1, which gives the surrogate for the log-determinant term:

$$\frac{1}{K} \sum_{k=1}^{K} \texttt{stop\_gradient}\left( \epsilon_k^\top f'(\hat{x}) \right) g'(z)\epsilon_k, \tag{37}$$

This is formulated in terms of the Jacobian of the decoder, in other words it is derived from $\det(J^\top J)$. The equivalent term, formulated in terms of the Jacobian of the encoder should be derived from $\det(J^\dagger J^{\dagger\top})$ and is therefore (note the negative sign):

$$-\frac{1}{K} \sum_{k=1}^{K} \epsilon_k^\top f'(\hat{x}) \texttt{stop\_gradient}\left( g'(z)\epsilon_k \right) \tag{38}$$

We write it in the order $f'(\hat{x})g'(z)$ rather than $g'(z)f'(\hat{x})$ since this reduces the variance of the estimate. See appendix D.1 for further details on this point.

### B.1 NOTE ON OPTIMALITY WITH RESPECT TO RECONSTRUCTION LOSS

Our estimator relies on the approximation $f'(\hat{x}) \approx g'(f(x))^\dagger$. If $f$ and $g$ are consistent ($f \circ g$ is the identity) this guarantees that $g'(f(x))f'(\hat{x}) = I$, but not that $f'(\hat{x})$ is the Moore-Penrose inverse of $g'(f(x))$. A sufficient requirement is that $f$ and $g$ are optimal with respect to the reconstruction loss, that is, any variation in the functions would lead to a higher reconstruction. With calculus of variations, it is possible to show that such $f$ and $g$ are consistent, and

$$(\hat{x} - x)^\top g'(f(x)) = 0 \tag{39}$$

for all $x$. By taking the derivative with respect to $x$ and evaluating at some $x$ in the image of $g$ (so $\hat{x} = x$) we have that

$$f'(\hat{x})^\top g'(f(x))^\top g'(f(x)) - g'(f(x)) = 0 \tag{40}$$

and hence

$$f'(\hat{x}) = g'(f(x))^\top (g'(f(x))^\top g'(f(x)))^{-1} = g'(f(x))^\dagger. \tag{41}$$

In the remainder of the appendix, given an encoder-decoder pair $f$ and $g$ which are optimal with respect to the reconstruction loss, we refer to $g$ as the pseudoinverse of $f$, and $f$ as the pseudoinverse of $g$.

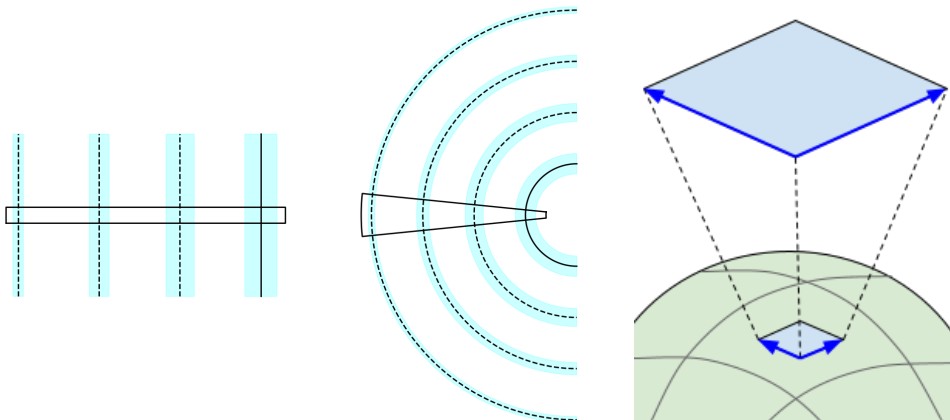

Figure 4: Representation of ill-defined probability density $\tilde{p}(x) \propto p(\hat{x})e^{-\beta\|\hat{x}-x\|^2}$ (*left* and *center*). Solid black lines denote the manifold, dashed lines are a constant distance from the manifold. The probability density is constant along the manifold. The width of the cyan bands is proportional to $e^{-\beta\|\hat{x}-x\|^2}$ and represents the probability density along the on- and off-manifold contours. While the density is reasonable for a flat manifold (*left*), note that the amount of probability mass associated with a region of the manifold (bounded by solid lines) is *larger* at some points off the manifold than on it when the manifold has curvature (*center*). This behavior can lead to divergent solutions when optimizing for likelihood and should be compensated for. The appropriate compensation factor is the ratio of the volume of a small region on the manifold (small blue square embedded in green manifold, *right*) to the equivalent region off the manifold (large blue square, *right*). The blue arrows represent an orthonormal frame on the manifold, and the equivalent frame in the off-manifold region.

## B.2 MODIFIED ESTIMATOR

As stated in 4.2, we modify the log-determinant estimator (eq. (38)) by replacing $f'(\hat{x})$ by $f'(x)$. The loss we are trying to optimize (sending gradient from the log-determinant to the encoder) is:

$$\mathcal{L}(x) = -\log p_Z(f(x)) - \log \mathrm{vol}\left(f'(\hat{x})\right) + \beta\|\hat{x} - x\|^2 \tag{42}$$

with $\mathrm{vol}(f'(\hat{x})) = \sqrt{\det(f'(\hat{x})f'(\hat{x})^\top)}$. Consider the probability density $\tilde{p}$ implied by interpreting this loss as a negative log-likelihood:

$$\tilde{p}(x) \propto p(\hat{x})e^{-\beta\|\hat{x}-x\|^2} = p_Z(z)\,\mathrm{vol}\left(f'(\hat{x})\right)e^{-\beta\|\hat{x}-x\|^2} \tag{43}$$

where $p(\hat{x})$ is the on-manifold density. Unfortunately, this density is ill-defined and leads to pathological behavior (see fig. 4). In order to provide a correction to this density, we need a term which compensates for the volume increase or decrease of off-manifold regions in comparison to the on-manifold region they are projected to. This is depicted in fig. 4 (*right*). The blue arrows in the on-manifold region will span the same latent-space volume as the blue arrows in the off-manifold region. The change in volume between the depicted on-manifold region and the latent space is $\mathrm{vol}(f'(\hat{x}))$ and $\mathrm{vol}(f'(x))$ between the off-manifold region and the latent space. Combining these facts means

$$\mathrm{vol}(f'(\hat{x}))\times(\text{volume of on-manifold region}) = \mathrm{vol}(f'(x))\times(\text{volume of off-manifold region}) \tag{44}$$

and hence the ratio of the volume of the on-manifold region to the off-manifold region is $\mathrm{vol}(f'(x))/\mathrm{vol}(f'(\hat{x}))$. Multiplying $\tilde{p}(x)$ by this factor leads to

$$\tilde{p}(x)\frac{\mathrm{vol}(f'(x))}{\mathrm{vol}(f'(\hat{x}))} = p_Z(z)\,\mathrm{vol}\left(f'(x)\right)e^{-\beta\|\hat{x}-x\|^2} \tag{45}$$

and the corresponding negative log-likelihood loss is

$$\mathcal{L}(x) = -\log p_Z(f(x)) - \log \mathrm{vol}\left(f'(x)\right) + \beta\|\hat{x} - x\|^2 \tag{46}$$

The surrogate for the log-determinant term is therefore

$$-\operatorname{tr}(f'(x)\texttt{stop\_gradient}(f'(x)^\dagger)) \tag{47}$$

In order to maintain computational efficiency, we approximate $f'(x)^\dagger$ by $g'(f(x))$:

$$-\operatorname{tr}(f'(x)\texttt{stop\_gradient}(g'(f(x)))) \tag{48}$$

giving the stated correction.

## C  LINEAR MODEL TRAINED ON MAXIMUM LIKELIHOOD ALONE

Consider a linear model, trained on data with zero mean and covariance $\Sigma$. Let the encoder function be $f(x) = Ax$ and suppose that $A$ has positive singular values, meaning that $AA^\top$ is positive definite. Let the decoder function be $g(z) = A^\dagger z$, where $A^\dagger = A^\top(AA^\top)^{-1}$. We want to minimize a combination of negative log-likelihood and a reconstruction loss (here we use $1/2\sigma^2$ instead of $\beta$ as prefactor):

$$\mathcal{L} = \mathcal{L}_{\text{NLL}} + \mathcal{L}_{\text{recon.}} \tag{49}$$

$$= E_x\left[\frac{1}{2}\|Ax\|^2 - \frac{1}{2}\log\det(AA^\top) + \frac{1}{2\sigma^2}\|A^\dagger Ax - x\|^2\right] \tag{50}$$

$$= \frac{1}{2}E_x[x^\top A^\top Ax] - \frac{1}{2}\log\det(AA^\top) + \frac{1}{2\sigma^2}E_x[x^\top(A^\dagger A - \mathbb{I})^2 x] \tag{51}$$

$$= \frac{1}{2}\operatorname{tr}(AE_x[xx^\top]A^\top) - \frac{1}{2}\log\det(AA^\top) + \frac{1}{2\sigma^2}\operatorname{tr}(E_x[xx^\top](\mathbb{I} - A^\dagger A)) \tag{52}$$

$$= \frac{1}{2}\operatorname{tr}(A\Sigma A^\top) - \frac{1}{2}\log\det(AA^\top) + \frac{1}{2\sigma^2}\operatorname{tr}(\Sigma(\mathbb{I} - A^\dagger A)) \tag{53}$$

where $A$ is a full rank $d \times D$ matrix with $d \leq D$.

Before solving for the minimum, let's review some matrix calculus identities. It is often convenient to consider $A$ as a function of a single variable $x$, differentiate with respect to $x$, and then choose $x$ to be $A_{ij}$. Then the derivative is

$$\frac{d}{dA_{ij}}A = E^{(ij)} \tag{54}$$

where $E^{(ij)}$ is a matrix of zeros, except for the $ij$ entry which is a one. We can write this as $E^{(ij)}_{kl} = \delta_{ik}\delta_{jl}$ where $\delta$ is the Kronecker delta. When evaluating $E^{(ij)}$ inside a trace we get the simple expression:

$$\operatorname{tr}(E^{(ij)}B) = E^{(ij)}_{kl}B_{lk} = \delta_{ik}\delta_{jk}B_{lk} = B_{ji} \tag{55}$$

using Einstein notation. The additional matrix identities we will need are Jacobi's formula for a square invertible matrix $B$:

$$\frac{d}{dx}\det(B) = \det(B)\operatorname{tr}\left(B^{-1}\left(\frac{d}{dx}B\right)\right) \tag{56}$$

and hence

$$\frac{d}{dx}\log\det(B) = \operatorname{tr}\left(B^{-1}\left(\frac{d}{dx}B\right)\right) \tag{57}$$

and we will prove the following lemma.

**Lemma C.1.** *Suppose the matrix $A$ depends on a variable $x$. Then we have the following expression for the derivative of the projection operator $A^\dagger A$:*

$$\frac{d}{dx}(A^\dagger A) = A^\dagger\left(\frac{d}{dx}A\right)(\mathbb{I} - A^\dagger A) + \left(A^\dagger\left(\frac{d}{dx}A\right)(\mathbb{I} - A^\dagger A)\right)^\top \tag{58}$$

*Proof.* The following is based on the proof to lemma 4.1 in Golub & Pereyra (1973). Define the projection operator $P_A = A^\dagger A$ and its complement $P_A^\perp = \mathbb{I} - P_A$. Then, since $P_A P_A = P_A$,

$$\left(\frac{d}{dx}P_A\right) = \left(\frac{d}{dx}P_A P_A\right) = \left(\frac{d}{dx}P_A\right)P_A + P_A\left(\frac{d}{dx}P_A\right) \tag{59}$$

In addition, since $P_A A^\top = A^\top$

$$\left( \frac{d}{dx} P_A A^\top \right) = \left( \frac{d}{dx} P_A \right) A^\top + P_A \left( \frac{d}{dx} A \right)^\top = \left( \frac{d}{dx} A \right)^\top \tag{60}$$

and therefore

$$\left( \frac{d}{dx} P_A \right) P_A = \left( \frac{d}{dx} P_A \right) A^\top A^{\dagger\top} = \left( \frac{d}{dx} A \right)^\top A^{\dagger\top} - P_A \left( \frac{d}{dx} A \right)^\top A^{\dagger\top} = P_A^\perp \left( \frac{d}{dx} A \right)^\top A^{\dagger\top} \tag{61}$$

By similar steps but using $A P_A = A$, we can derive

$$P_A \left( \frac{d}{dx} P_A \right) = A^\dagger \left( \frac{d}{dx} A \right) P_A^\perp \tag{62}$$

Putting it all together gives

$$\left( \frac{d}{dx} P_A \right) = A^\dagger \left( \frac{d}{dx} A \right) P_A^\perp + \left( A^\dagger \left( \frac{d}{dx} A \right) P_A^\perp \right)^\top \tag{63}$$

Note that the second term is just the transpose of the first. $\qquad\square$

Now we are ready to find the derivative of the loss and set it to zero.

**Lemma C.2.** *The derivative of the loss with respect to $A$ takes the form:*

$$\frac{d}{dA} \mathcal{L} = \left( \Sigma A^\top - A^\dagger - \frac{1}{\sigma^2} (\mathbb{I} - A^\dagger A) \Sigma A^\dagger \right)^\top \tag{64}$$

*Proof.* Let's apply the above identities to the first term in the loss:

$$\frac{d}{dx} \frac{1}{2} \operatorname{tr}(A \Sigma A^\top) = \frac{1}{2} \operatorname{tr} \left( \left( \frac{d}{dx} A \right) \Sigma A^\top + A \Sigma \left( \frac{d}{dx} A \right)^\top \right) \tag{65}$$

$$= \operatorname{tr} \left( \left( \frac{d}{dx} A \right) \Sigma A^\top \right) \tag{66}$$

since the trace is invariant under transposition and hence

$$\frac{d}{dA_{ij}} \frac{1}{2} \operatorname{tr}(A \Sigma A^\top) = \operatorname{tr}(E^{(ij)} \Sigma A^\top) = (\Sigma A^\top)_{ji} \tag{67}$$

Applying Jacobi's formula to the second term in the loss gives:

$$\frac{d}{dx} \frac{1}{2} \log \det(A A^\top) = \frac{1}{2} \operatorname{tr} \left( (A A^\top)^{-1} \left( \frac{d}{dx} (A A^\top) \right) \right) \tag{68}$$

$$= \frac{1}{2} \operatorname{tr} \left( (A A^\top)^{-1} \left( \left( \frac{d}{dx} A \right) A^\top + A \left( \frac{d}{dx} A \right)^\top \right) \right) \tag{69}$$

and therefore

$$\frac{d}{dA_{ij}} \frac{1}{2} \log \det(A A^\top) = \frac{1}{2} \operatorname{tr} \left( (A A^\top)^{-1} \left( E^{(ij)} A^\top + A E^{(ji)} \right) \right) \tag{70}$$

$$= \frac{1}{2} \operatorname{tr} \left( E^{(ij)} A^\top (A A^\top)^{-1} + E^{(ij)} A^\top (A A^\top)^{-1} \right) \tag{71}$$

$$= \left( A^\top (A A^\top)^{-1} \right)_{ji} \tag{72}$$

$$= A^\dagger_{ji} \tag{73}$$

where we used the cyclic and transpose properties of the trace and that $E^{(ji)\top} = E^{(ij)}$.

The final term requires a derivative of $\mathrm{tr}(\Sigma(\mathbb{I} - A^\dagger A))$, which is equal to a derivative of $-\mathrm{tr}(\Sigma A^\dagger A)$. We use the formula for the derivative of the projection operator to get

$$\frac{d}{dx} \frac{1}{2} \mathrm{tr}(\Sigma A^\dagger A) = \frac{1}{2} \mathrm{tr}\left(\Sigma \left(\frac{d}{dx}(A^\dagger A)\right)\right) \tag{74}$$

$$= \mathrm{tr}\left(\Sigma A^\dagger \left(\frac{d}{dx} A\right)(\mathbb{I} - A^\dagger A)\right) \tag{75}$$

again using the transpose property of the trace, and therefore

$$\frac{d}{dA_{ij}} \frac{1}{2} \mathrm{tr}(\Sigma A^\dagger A) = \mathrm{tr}(E^{(ij)}(\mathbb{I} - A^\dagger A)\Sigma A^\dagger) = ((\mathbb{I} - A^\dagger A)\Sigma A^\dagger)_{ji} \tag{76}$$

Putting the three expressions together, we have that

$$\frac{d}{dA}\mathcal{L} = \left(\Sigma A^\top - A^\dagger - \frac{1}{\sigma^2}(\mathbb{I} - A^\dagger A)\Sigma A^\dagger\right)^\top \tag{77}$$

$\square$

**Lemma C.3.** *The critical points of $\mathcal{L}$ satisfy the following properties:*

1. $A = U\Sigma^{-\frac{1}{2}}$ with $UU^\top = \mathbb{I}$

2. $U^\top U$ commutes with $\Sigma$

*Proof.* Using lemma C.2, the critical points satisfy

$$\Sigma A^\top - A^\dagger - \frac{1}{\sigma^2}(\mathbb{I} - A^\dagger A)\Sigma A^\dagger = 0 \tag{78}$$

By multiplying by $A$ from the left we have

$$A\Sigma A^\top = \mathbb{I}_d \tag{79}$$

meaning that $U = A\Sigma^{\frac{1}{2}}$ must have orthonormal rows (since $UU^\top = \mathbb{I}$). With this definition, we can write $A = U\Sigma^{-\frac{1}{2}}$.

If we now multiply by $A$ from the right, we get

$$\Sigma A^\top A - A^\dagger A - \frac{1}{\sigma^2}\Sigma A^\dagger A + \frac{1}{\sigma^2}A^\dagger A\Sigma A^\dagger A = 0 \tag{80}$$

Noting that the second and fourth terms are symmetric (since $A^\dagger A$ is symmetric), this means that the remaining terms must be symmetric:

$$\Sigma A^\top A - \frac{1}{\sigma^2}\Sigma A^\dagger A = A^\top A\Sigma - \frac{1}{\sigma^2}A^\dagger A\Sigma \tag{81}$$

Since $A^\top A$ commutes with $A^\dagger A$, they are simultaneously diagonalizable, and since they are both symmetric, they share an orthonormal basis of eigenvectors. Clearly $A^\top A - A^\dagger A/\sigma^2$ has the same basis. Since this matrix commutes with $\Sigma$, it must share a basis with $\Sigma$ and hence $\Sigma$ has the same basis as $A^\top A$. This means that $\Sigma$ commutes with $A^\top A$.

Expanding $A$ in terms of $U$, this means that

$$\Sigma A^\top A = \Sigma^{\frac{1}{2}} U^\top U\Sigma^{-\frac{1}{2}} = \Sigma^{-\frac{1}{2}} U^\top U\Sigma^{\frac{1}{2}} = A^\top A\Sigma \tag{82}$$

and therefore $\Sigma U^\top U = U^\top U\Sigma$, meaning that $\Sigma$ and $U^\top U$ commute. $\square$

Consider as an example the case where $\Sigma$ is diagonal. $U^\top U$ is a projection matrix and in this case must be diagonal due to commuting with $\Sigma$. As a result, it must have exactly $d$ ones and $D - d$ zeros along the diagonal. This means that the rows of $U$ are a basis for the $d$ dimensional axis-aligned subspace corresponding to the $d$ nonzero entries. In the case of a non-diagonal $\Sigma$, this generalizes to the rows of $U$ spanning the same subspace as some subset of $d$ eigenvectors of $\Sigma$. This leads to the expression of the loss function in the next theorem.

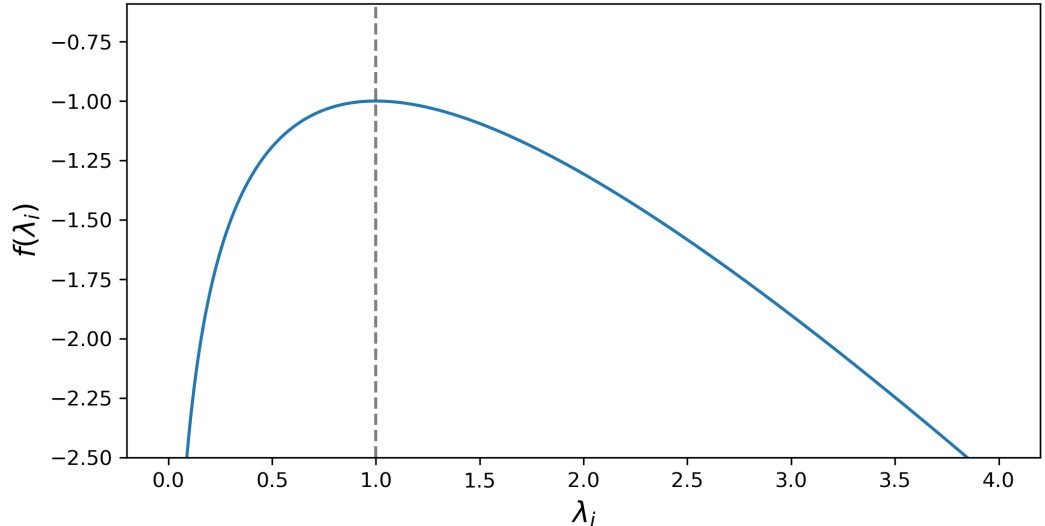

Figure 5: Plot of $f(\lambda_i) = \log \lambda_i - \lambda_i / \sigma^2$ with $\sigma = 1$, showing maximum value at $\lambda_i = \sigma^2$ and unbounded behavior on either side.

**Theorem C.4.** *Let $\Sigma$ have the eigen-decomposition $V \Lambda V^\top$ with $\Lambda = \mathrm{diag}(\lambda)$. Let $U^\top U$ have the eigen-decomposition $V E V^\top$, with $E = \mathrm{diag}(\alpha)$. Then the minimum of the loss is satisfied by $\alpha$ such that*

$$\mathcal{L}_\alpha = \sum_{i=1}^{D} \frac{1}{2} \alpha_i \left( \log \lambda_i - \frac{1}{\sigma^2} \lambda_i \right) \tag{83}$$

*is minimal, subject to the constraint $\alpha_i \in \{0, 1\}$ with $\sum_{i=1}^{D} \alpha_i = d$.*

*Proof.* Let's note a couple of properties. First, we have $(U \Sigma U^\top)^k = U \Sigma^k U^\top$ due to $\Sigma$ commuting with $U^\top U$, so we can say that $f(U \Sigma U^T) = U f(\Sigma) U^\top$ for any matrix function $f$ with a Taylor series. Next, $U^\top U$ is an orthogonal projection matrix, so $E$ is a diagonal matrix with ones or zeros on the diagonal. We know that the rank of $U$ is $d$, hence $E$ has exactly $d$ ones and $D - d$ zeros along the diagonal. Therefore we have the constraint $\alpha_i \in \{0, 1\}$ with $\sum_{i=1}^{D} \alpha_i = d$. Next, note that $A^\dagger A = U^\top U$.

Now we substitute back into the loss in terms of $U$:

$$\mathcal{L} = \frac{1}{2} \mathrm{tr}(U U^\top) - \frac{1}{2} \log \det(U \Sigma^{-1} U^\top) + \frac{1}{2\sigma^2} \mathrm{tr}(\Sigma(\mathbb{I} - U^\top U)) \tag{84}$$

$$= \frac{1}{2} \mathrm{tr}(U \log(\Sigma) U^\top) - \frac{1}{2\sigma^2} \mathrm{tr}(U \Sigma U^\top) + \mathrm{const.} \tag{85}$$

where we used that

$$\log \det(U \Sigma^{-1} U^\top) = \mathrm{tr} \log(U \Sigma^{-1} U^\top) = - \mathrm{tr}(U \log(\Sigma) U^\top) \tag{86}$$

Note that $\mathrm{tr}(U U^\top)$ is constant. Consider that

$$\mathrm{tr}(U \Sigma U^\top) = \mathrm{tr}(U^\top U \Sigma U^\top U) = \mathrm{tr}(V E D E V^\top) = \mathrm{tr}(E D E) = \alpha \cdot \lambda \tag{87}$$

The same logic holds for the term with $\log(\Sigma)$. Therefore, dropping constant terms, the loss can be written in terms of $\alpha$ and $\lambda$:

$$\mathcal{L}_\alpha = \sum_{i=1}^{D} \frac{1}{2} \alpha_i \left( \log \lambda_i - \frac{1}{\sigma^2} \lambda_i \right) \tag{88}$$

$\square$

The loss will take different values depending on which elements of $\alpha$ are nonzero. Define $f(\lambda_i) = \log \lambda_i - \lambda_i/\sigma^2$. The loss will be minimized when the nonzero $\alpha_i$ correspond to those values of $f(\lambda_i)$ which are minimal. Clearly $f''(\lambda_i) < 0$, so $f$ has only one maximum at $\lambda_i = \sigma^2$ and is unbounded below on either side of this maximum (see fig. 5). Consider the two extreme cases:

1. All eigenvalues $\lambda$ are smaller than $\sigma^2$. The minimal values of $f(\lambda_i)$ will occur for the smallest values of $\lambda_i$. Hence the $d$ smallest eigenvalues of $\Sigma$ will be selected.

2. All eigenvalues $\lambda$ are larger than $\sigma^2$. The minimal values of $f(\lambda_i)$ will occur for the largest values of $\lambda_i$. Hence the $d$ largest eigenvalues of $\Sigma$ will be selected.

In the intermediate regime, there will be a phase transition between these two extremes.

In the first case, the reconstructed manifold will be a projection onto the $d$-dimensional subspace with the lowest variance, exactly the opposite result to PCA. In the second case, the reconstructed manifold will be a projection onto the $d$-dimensional subspace with the highest variance, exactly the same result as PCA. If we maximize the likelihood on the manifold without any reconstruction loss, corresponding to the $\sigma^2 \to \infty$ limit, we actually learn the lowest entropy manifold. It makes more sense to learn the highest entropy manifold as in PCA. We can ensure this is the case by adding Gaussian noise of variance $\sigma^2$ to the data, ensuring that the minimum eigenvalue of the covariance matrix is at least $\sigma^2$, even if the original data is degenerate.

## D  IMPLEMENTATION DETAILS

In implementing the trace estimator, we have to make a number of choices, each elaborated below. The main reasons for each choice are given first, with more technical details deferred to later in the appendix.

**Gradient to encoder or decoder**  The log-determinant term can be formulated either in terms of the Jacobian of the encoder (see eq. (10)) or the decoder (see eq. (9) ). As discussed in section 4.2 we find that formulating it in terms of the encoder Jacobian leads to more stable training. Since the training also minimizes the squared norm of $f(x)$, we speculate that having gradient from this term and the log-determinant term both being sent to the encoder allows the encoder to more efficiently shape the latent space distribution. We note the similarity of this formulation to the standard change-of-variables loss used to train normalizing flows. If we instead send the gradient of the log-determinant to the decoder, the information about how the encoder can change can only reach it via the reconstruction term, which doesn't allow the encoder to deviate significantly from being the pseudoinverse of the decoder. A change in the decoder will therefore lead to a corresponding change in the encoder, but this is a less direct process than sending gradient to the encoder directly. In addition, this formulation means the decoder is optimized only to minimize reconstruction loss, meaning that it will likely be an approximate pseudoinverse for the encoder, a condition we require for the accuracy of the surrogate estimator.

**Space in which trace is performed**  Considering eqs. (29) and (36), the central component of the surrogate is a trace (estimator). Making use of the cyclic property of the trace, i.e. $\text{tr}(A^\top B) = \text{tr}(BA^\top)$ for any $A, B \in \mathbb{R}^{D \times d}$, we can choose which expansion of the trace to estimate:

$$\sum_{i=1}^{d}(A^\top B)_{ii} = \text{tr}(A^\top B) = \text{tr}(BA^\top) = \sum_{i=1}^{D}(BA^\top)_{ii}. \tag{89}$$

The variance of a stochastic trace estimator depends on the noise used but in general is roughly proportional to the squared Frobenius norm of the matrix (see appendix D.3). Given two matrices $A, B \in \mathbb{R}^{D \times d}$ with $d < D$, it is likely that $\|A^\top B\|_F^2 < \|BA^\top\|_F^2$. This statement is not true for all $A$ and $B$, but is almost always fulfilled when $d \ll D$.

Transferred to our context: In general the matrices $f'(x) \in \mathbb{R}^{d \times D}$ and $g'(z) \in \mathbb{R}^{D \times d}$ are rectangular and can be multiplied together in either the $f'(x)g'(z) \in \mathbb{R}^{d \times d}$ order or $g'(z)f'(x) \in \mathbb{R}^{D \times D}$ order. This matters for applying the trace since generally $d < D$.

A more precise statement (proven in appendix D.1) is that if the entries of $A$ and $B$ are sampled from standard normal distributions, then $E[\|A^\top B\|_F^2] = Dd^2$ versus $E[\|BA^\top\|_F^2] = D^2d$. For

Table 4: Different possible estimators for the gradient of the log-determinant term.

|  | gradient to encoder | gradient to decoder |
|---|---|---|
| trace in data space | $-\operatorname{tr}\left(g'(z)\left(\frac{\partial}{\partial \phi_j} f'(x)\right)\right)$ | $\operatorname{tr}\left(\left(\frac{\partial}{\partial \theta_j} g'(x)\right) f'(x)\right)$ |
| trace in latent space | $-\operatorname{tr}\left(\left(\frac{\partial}{\partial \phi_j} f'(x)\right) g'(z)\right)$ | $\operatorname{tr}\left(f'(x)\left(\frac{\partial}{\partial \theta_j} g'(z)\right)\right)$ |

$d \ll D$ the difference becomes significant. The difference between the two estimators may not be large if the two matrices have special structure, in particular if they share a basis. However, since the terms being multiplied in our case are a Jacobian matrix and the derivative of another Jacobian matrix with respect to a parameter $\theta_j$ or $\phi_j$, it is unlikely that any such particular structure is present.

As a result, when the latent space is smaller than the data space, the preferable estimator is the one that performs the trace in the latent space, meaning that products in the estimator have the order $f'(x)g'(z)$ (see table 4). In appendix D.1.1, we experimentally test the convergence of trace estimators with increasing Hutchinson samples, performed in both data and latent space, confirming that convergence is much faster when performing the trace in latent space.

**Type of gradient**  Consider the estimator:

$$\operatorname{tr}\left(\left(\frac{\partial}{\partial \phi_j} f'(x)\right) g'(z)\right) = \frac{\partial}{\partial \phi_j} E_\epsilon\left[\epsilon^\top f'(x) \texttt{stop\_gradient}(g'(z))\epsilon\right] \tag{90}$$

Ignoring the stop gradient operation for now, this requires computing terms of the form $\epsilon^\top f'(x)g'(z)\epsilon$. In order to avoid calculating full Jacobian matrices, we can implement the calculation using some combination of vector-Jacobian (`vjp`) or Jacobian-vector (`jvp`) products, which are efficient to compute with backward-mode respectively forward-mode automatic differentiation. Note that we can use the result from one product as the vector for another `vjp` or `jvp`. For example, $v_1 := (\epsilon^\top f'(x))^\top \in \mathbb{R}^D$ yields a vector, so we can compute $\epsilon^\top f'(x)g'(z)\epsilon = v_1^\top g'(z)\epsilon$ via two vector-Jacobian products.

This gives us three choices: i) backward mode only (two `vjp`), ii) forward mode only (two `jvp`) or iii) a mix of both (one `jvp` and one `jvp`). We opt to use mixed mode (see appendix D.2 for further details).

**Trace estimator noise**  Trace estimators rely on the identity $E_\epsilon[\epsilon^\top A\epsilon] = \operatorname{tr}(AE_\epsilon[\epsilon\epsilon^\top]) = \operatorname{tr}(A)$, meaning that we require only $E_\epsilon[\epsilon\epsilon^\top] = \mathbb{I}$ for the noise variable. The choice comes down mainly to the variance of the estimator. Among all noise vectors whose entries are sampled independently, Rademacher noise has the lowest variance (Hutchinson, 1989). However, if the entries are sampled from a standard normal distribution and then scaled to have length $\sqrt{d}$ where $d$ is the dimension of $\epsilon$, the entries are no longer independent and the variance of the estimator is comparable to Rademacher noise (Girard, 1989). When using a single Hutchinson sample, we choose to use scaled Gaussian noise for its low variance, and since it covers more directions than Rademacher noise (covering the hypersphere uniformly, rather than at a fixed $2^d$ points). When we have more than one Hutchinson sample, we additionally orthogonalize the vectors as this further reduces variance. More details are in appendix D.3.

**Number of noise samples**  We can choose to use between 1 and $d$ noise samples in the trace estimator (with $d$ samples we already can calculate the exact trace, so more samples are not necessary). Denote the number of samples by $K$. We find that in general $K = 1$ is enough for good performance, especially if the batch size is sufficiently high.

## D.1  VARIANCE OF TRACE ESTIMATOR

**Theorem D.1.** *Let $A, B \in \mathbb{R}^{D \times d}$ where the entries of both matrices are sampled from a standard normal distribution. Then*

$$E\left[\|A^\top B\|_F^2\right] = d^2 D \qquad and \qquad E\left[\|BA^\top\|_F^2\right] = dD^2 \tag{91}$$

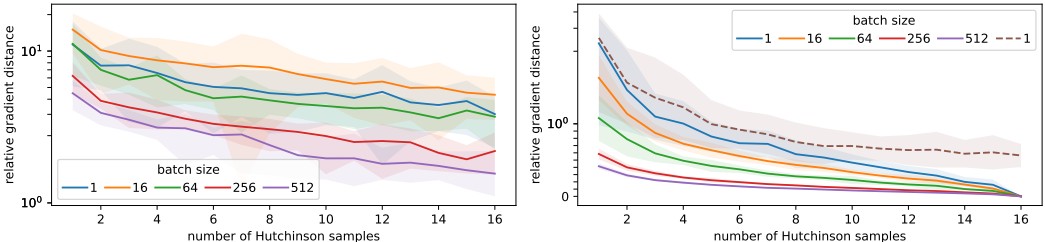

Figure 6: Relative gradient distance to the exact surrogate gradient as a function of Hutchinson samples for varying batch sizes. *(Left)* Trace estimation in data space. *(Right)* Trace estimation in latent space. We estimate the trace using orthogonalized Gaussian noise (see appendix D.3.5) (solid lines), but also feature non-orthogonalized samples for batch size 1 in latent space (dashed line). Note the different scales on the y-axes and the use of `symlog` in the right-hand plot. We evaluate the trace estimation on the surrogate estimator with gradient to encoder as specified in table 4 for a converged model trained on conditional MNIST ($d = 16$, $D = 784$).

*Proof.* Consider first $\|A^\top B\|_F^2$. We can write this as

$$\|A^\top B\|_F^2 = \sum_{i=1}^{d} \sum_{j=1}^{d} \left( \sum_{k=1}^{D} A_{ki} B_{kj} \right)^2 = \sum_{i,j}^{d} \sum_{k,l}^{D} A_{ki} A_{li} B_{kj} B_{lj} \tag{92}$$

Taking an expectation over this expression, the only nonzero contributions will be from terms where the $A$ and $B$ terms are both quadratic, since if not, the term will be multiplied by $E[X] = 0$ where $X$ is standard normal. This requires $k = l$, giving

$$E\left[\|A^\top B\|_F^2\right] = \sum_{i,j}^{d} \sum_{k}^{D} E\left[A_{ki}^2 B_{kj}^2\right] \tag{93}$$

$$= \sum_{i,j}^{d} \sum_{k}^{D} E\left[A_{ki}^2\right] E\left[B_{kj}^2\right] \tag{94}$$

$$= d^2 D \tag{95}$$

since $A_{ki}$ and $B_{kj}$ are independent and the expectation of the square of a standard normal variable is its variance, i.e. 1. The equivalent expressions for $\|BA^\top\|_F^2$ can be obtained by swapping $d$ and $D$ in these expressions. □

### D.1.1 EXPERIMENTAL CONFIRMATION

To evaluate the convergence behavior of the trace estimation, which is exact for $d$ and $D$ Hutchinson samples in latent and data space respectively, we compute the relative gradient distance of the resulting surrogate gradient with respect to the exact solution as a function of Hutchinson samples $K$:

$$\text{relative gradient distance}(K) = \frac{\|\nabla \text{surrogate}(K) - \nabla \text{exact}\|_2}{\|\nabla \text{exact}\|_2}. \tag{96}$$

Here, $\nabla \text{surrogate}(K)$ denotes the gradient of the surrogate loss term after $K$ Hutchinson samples and $\nabla \text{exact}$ the gradient of the exact surrogate loss term, i.e. after $d$ or $D$ samples.

In fig. 6 one can see a clear decrease in gradient distance and its variance when computing the trace in latent space instead of data space. Furthermore, we note that increasing the batch size also contributes to fast and steady convergence, which is a result of sampling an independent noise sample per batch instance.

### D.2 FORWARD/BACKWARD AUTOMATIC DIFFERENTIATION

Two of the basic building blocks of automatic differentiation (autodiff) libraries are the vector-Jacobian product (`vjp`) and the Jacobian-vector product (`jvp`). The `vjp` is implemented by

backward-mode autodiff and computes a vector multiplied with a Jacobian matrix from the left, along with the output of the function being used:

$$f(x), \ \epsilon^\top f'(x) = \mathtt{vjp}(f, x, \epsilon) \tag{97}$$

In PyTorch, this is implemented by the `torch.autograd.functional.vjp` function, or by first computing $f(x)$, then using `torch.autograd.grad`.

The `jvp` is implemented by forward-mode autodiff and computes a vector multiplied by a Jacobian matrix from the right, along with the output of the function being used:

$$f(x), \ f'(x)\epsilon = \mathtt{jvp}(f, x, \epsilon) \tag{98}$$

In PyTorch, this is implemented by the `torch.autograd.forward_ad` package.

Our preferred estimator for the log-determinant is the following (see section 4.2):

$$-\frac{1}{K}\sum_{k=1}^{K} \epsilon_k^\top f'(x)\mathtt{stop\_gradient}\left(g'(z)\epsilon_k\right) \tag{99}$$

Therefore we need to compute terms of the form $\epsilon^\top f'(x)g'(z)\epsilon$, with a `stop_gradient` operation. The `stop_gradient` operation is implemented by applying the `.detach()` method to a tensor in PyTorch. We have the following options. Note that we can use a product obtained from `vjp` or `jvp` as a vector input to a subsequent product.

**Backward mode**   This mode uses only vector-Jacobian products, requiring backward-mode autodiff.

1. $v_1^\top = \epsilon^\top f'(x)$ $\qquad\qquad\qquad\qquad\qquad\qquad\qquad\quad$ $z, \ v_1^\top = \mathtt{vjp}(f, x, \epsilon)$
2. $v_2^\top = v_1^\top g'(z)$ $\qquad\qquad\qquad\qquad\qquad\qquad\qquad\quad$ $\hat{x}, \ v_2^\top = \mathtt{vjp}(g, z, v_1)$
3. $\epsilon^\top f'(x)g'(z)\epsilon = v_2^\top \epsilon$

**Forward mode**   This mode uses only Jacobian-vector products, requiring forward-mode autodiff.

1. $v_1 = g'(z)\epsilon$ $\qquad\qquad\qquad\qquad\qquad\qquad\qquad\qquad\quad$ $\hat{x}, \ v_1 = \mathtt{jvp}(g, z, \epsilon)$
2. $v_2 = f'(x)v_1$ $\qquad\qquad\qquad\qquad\qquad\qquad\qquad\qquad\quad$ $z, \ v_2 = \mathtt{jvp}(f, x, v_1)$
3. $\epsilon^\top f'(x)g'(z)\epsilon = \epsilon^\top v_2$

**Mixed mode**   This mode uses one vector-Jacobian and one Jacobian-vector product, requiring both forward- and backward-mode autodiff.

1. $v_1^\top = \epsilon^\top f'(x)$ $\qquad\qquad\qquad\qquad\qquad\qquad\qquad\quad$ $z, \ v_1 = \mathtt{vjp}(f, x, \epsilon)$
2. $v_2 = g'(z)\epsilon$ $\qquad\qquad\qquad\qquad\qquad\qquad\qquad\qquad\quad$ $\hat{x}, \ v_2 = \mathtt{jvp}(g, z, \epsilon)$
3. $\epsilon^\top f'(x)g'(z)\epsilon = v_1^\top v_2$

We prefer using backward mode autodiff where possible, since we find that it is slightly faster than forward mode in PyTorch. However for our estimator of choice, we use mixed mode, since this is most easily implemented. Using backward mode would require a `stop_gradient` operation to be introduced after the second step, but in a way that allows gradient to flow back to $f'(x)$. While we believe this is possible if implemented carefully, we did not pursue this option. In mixed mode, we can easily detach the gradient of $v_2$ without affecting the first step of the calculation.

### D.3   PROPERTIES OF TRACE ESTIMATOR NOISE

Hutchinson style trace estimators (Hutchinson, 1989) have the form $E_\epsilon[\epsilon^\top A\epsilon]$ and equal $\mathrm{tr}(A)$ in expectation. If $A$ is skew-symmetric ($A^\top = -A$), then $(\epsilon^\top A\epsilon)^\top = \epsilon^\top A^\top \epsilon = -\epsilon^\top A\epsilon$ and hence $\epsilon^\top A\epsilon = 0$ with zero variance. Since any matrix can be decomposed into a symmetric and skew-symmetric part, the variance in the estimator comes only from the symmetric part of $A$, namely $A_s = (A + A^\top)/2$. From now on, suppose $A$ is symmetric and if not, substitute $A_s$ for $A$.

### D.3.1 RADEMACHER NOISE

If the entries of $\epsilon$ are sampled independently from a distribution with zero mean and unit variance, then the variance of the estimator is minimized by the Rademacher distribution which samples the values $-1$ and $1$ each with probability half. This estimator achieves the following variance for symmetric $A$ (see proposition 1 in Hutchinson (1989)):

$$V_\epsilon[\epsilon^\top A\epsilon] = 2\sum_{i\neq j} A_{ij}^2 \tag{100}$$

### D.3.2 GAUSSIAN NOISE

With standard normal noise, the estimator is unbiased, but the variance is higher (see again Hutchinson (1989)):

$$V_\epsilon[\epsilon^\top A\epsilon] = 2\sum_{i,j} A_{ij}^2 = 2\|A\|_F^2 \tag{101}$$

i.e. twice the Frobenius norm.

### D.3.3 SCALED GAUSSIAN NOISE

By contrast

$$E_\epsilon\left[\frac{\epsilon^\top A\epsilon}{\epsilon^\top \epsilon}\right] = \frac{1}{d}\operatorname{tr}(A) \tag{102}$$

where $\epsilon$ is a standard normal variable in $\mathbb{R}^d$. The variance of this estimator for symmetric $A$ (see theorem 2.2 in Girard (1989)) is:

$$V_\epsilon\left[\frac{\epsilon^\top A\epsilon}{\epsilon^\top \epsilon}\right] = \frac{2}{d+2}\sigma^2(\lambda(A)) \tag{103}$$

where $\sigma^2(\lambda(A))$ denotes the variance of the eigenvalues of $A$.

We can write this estimator in the "Hutchinson" form by sampling $\epsilon$ from a standard normal distribution, then normalizing it such that its length is $\sqrt{d}$. Then we have

$$E_\epsilon[\epsilon^\top A\epsilon] = \operatorname{tr}(A) \tag{104}$$

and

$$V_\epsilon[\epsilon^\top A\epsilon] = \frac{2d^2}{d+2}\sigma^2(\lambda(A)) \tag{105}$$

### D.3.4 COMPARISON

When the dimension of $A$ becomes large, the variance of Rademacher and scaled Gaussian estimators are comparable. Suppose that the eigenvalues of $A$ have zero mean (e.g. the entries are independent normal samples). Then

$$d\sigma^2(\lambda(A)) = \sum_i \lambda_i^2 = \operatorname{tr}(A^2) = \|A\|_F^2 \tag{106}$$

If we further assume that all entries of $A$ have roughly equal magnitude we have that

$$\sum_{i\neq j} A_{ij}^2 \approx \|A\|_F^2 \tag{107}$$

since the sum in the Frobenius norm is dominated by the $d(d-1)$ off-diagonal terms. Similarly,

$$\frac{d^2}{d+2}\sigma^2(\lambda(A)) \approx d\sigma^2(\lambda(A)) = \|A\|_F^2 \tag{108}$$

meaning that the two estimators have approximately the same variance.

If the matrix has special structure, we might choose one estimator over the other. For example, if the standard deviation of the eigenvalues of $A$ is small in comparison to the mean eigenvalue, the

scaled Gaussian estimator is preferable and if $A$ is dominated by its diagonal then the Rademacher estimator is preferable. We don't expect either type of special structure in our matrices, so we consider the estimators interchangeable. We decided to use scaled Gaussian noise since it produces noise which points in all possible directions in $\mathbb{R}^d$ whereas Rademacher noise is restricted to a finite $2^d$ points. We assume that there is no reason to prefer this set of $2^d$ directions and therefore sampling from all possible directions is better.

### D.3.5 REDUCING VARIANCE WHEN SAMPLING MORE THAN 1 HUTCHINSON SAMPLE

When the number of Hutchinson samples $K$ are greater than 1, it is more favorable to sample the noise vectors in a dependent way than independently. Consider the case of a $d \times d$ matrix $A$ with $K = d$. Then we can get an exact estimate of the trace via

$$\sum_{i=1}^{d} q_i^\top A q_i = \text{tr}(Q^\top A Q) = \text{tr}(A) \tag{109}$$

with orthogonal $Q$ and $q_i$ the $i$-th column of $Q$. If the $q_i$ were sampled independently, we almost certainly wouldn't achieve this exact result. We therefore sample our noise vectors as the first $K$ columns of a randomly sampled $d \times d$ orthogonal matrix and scale each column by $\sqrt{d}$. We show below that this reduces variance compared with sampling independently and make an experimental comparison (see fig. 6). If the resulting noise vectors are denoted $\epsilon_i$, we estimate $\text{tr}(A)$ by

$$\widehat{\text{tr}}(A) = \frac{1}{K} \sum_{i=1}^{K} \epsilon_i^\top A \epsilon_i \tag{110}$$

This estimator is unbiased:

$$E_{\epsilon_1,\dots,\epsilon_K}[\widehat{\text{tr}}(A)] = \frac{1}{K} \sum_{i=1}^{K} E_{\epsilon_i}[\epsilon_i^\top A \epsilon_i] \tag{111}$$

$$= \frac{1}{K} \sum_{i=1}^{K} \text{tr}\left(A E_{\epsilon_i}[\epsilon_i \epsilon_i^\top]\right) \tag{112}$$

$$= \frac{1}{K} \sum_{i=1}^{K} \text{tr}(A) \tag{113}$$

$$= \text{tr}(A) \tag{114}$$

since $E_{\epsilon_i}[\epsilon_i \epsilon_i^\top] = \mathbb{I}$ for all $\epsilon_i$.

This procedure is equivalent to using scaled Gaussian noise when $K = 1$ and is what we implement in practice for all values of $K$. Note that it is not necessary to use $K > d$ since we already achieve the exact value with $K = d$.

A note on our sampling strategy: in practice we sample by taking the $Q$ matrix of the QR decomposition of a $d \times K$ matrix with entries sampled from a standard normal. Since the QR decomposition performs Gram-Schmidt orthogonalization, the $Q$ matrix is uniformly sampled from the group of orthogonal matrices if $Q$ is square (Mezzadri, 2007). The same logic applies to the QR decomposition of non-square matrices, yielding a $d \times K$ matrix made up of the first $K$ columns of a uniformly sampled orthogonal $d \times d$ matrix. Strictly speaking, the QR decomposition is only unique if the $R$ matrix has a positive diagonal, and uniqueness is required for uniform sampling. Let $X = QR$ and define by $D$ the sign of the diagonal of $R$. $D$ is diagonal with 1 or $-1$ on the diagonal and $D^2 = \mathbb{I}$. Uniqueness can be achieved by multiplying $Q$ by $D$ from the right and multiplying $R$ by $D$ from the left: $X = QDDR = QR$. The resulting uniformly sampled orthogonal matrix is $QD$, meaning the columns of $Q$ are multiplied by either 1 or $-1$. In our setting, we have terms of the form $\epsilon_i^\top A \epsilon_i$ where the $\epsilon_i$ are the columns of $Q$, so multiplying the columns by $-1$ has no effect on the trace estimate. As a result we opt not to multiply by $D$. Therefore, although we do not sample uniformly from the orthogonal group, the final result is equivalent to sampling uniformly.

**Variance derivation**  Using the formula for the variance of a sum of random variables, we have that

$$V_{\epsilon_1,\dots,\epsilon_K}[\widehat{\mathrm{tr}}(A)] = \frac{1}{K^2} \sum_{i,j=1}^{K} C_{\epsilon_i,\epsilon_j}[\epsilon_i^\top A\epsilon_i, \epsilon_j^\top A\epsilon_j] \tag{115}$$

where $C$ denotes the covariance between two random variables. Note that since the permutation of the columns of a randomly sampled orthogonal matrix is arbitrary (permuting the columns results in another randomly sampled orthogonal matrix of equal probability), all columns are equivalent and we only have to distinguish between the cases $i = j$ and $i \neq j$. This leads to[1]

$$V_{\epsilon_1,\dots,\epsilon_K}[\widehat{\mathrm{tr}}(A)] = \frac{1}{K^2}(Kv + K(K-1)c) = \frac{1}{K}(v + (K-1)c) \tag{116}$$

with $v = V_{\epsilon_i}[\epsilon_i^\top A\epsilon_i]$ and $c = C_{\epsilon_i,\epsilon_j}[\epsilon_i^\top A\epsilon_i, \epsilon_j^\top A\epsilon_j]$ when $i \neq j$. Each column viewed individually is a randomly sampled Gaussian vector, scaled to have length $\sqrt{d}$, hence the value of $v$ is equal to the scaled Gaussian noise case above, namely

$$v = \frac{2d^2}{d+2}\sigma^2(\lambda(A)) \tag{117}$$

where $\sigma^2(\lambda(A))$ denotes the variance of the eigenvalues of $A$. We also know that the variance of the estimator reduces to zero when $K = d$ and hence $v + (d-1)c = 0$ leading to

$$c = -\frac{v}{d-1} \tag{118}$$

Putting it all together leads to

$$V_{\epsilon_1,\dots,\epsilon_K}[\widehat{\mathrm{tr}}(A)] = \frac{1}{K}(v - \frac{K-1}{d-1}v) \tag{119}$$

$$= \frac{1}{K}\frac{d-K}{d-1}v \tag{120}$$

$$= \frac{2d^2(d-K)}{K(d-1)(d+2)}\sigma^2(\lambda(A)) \tag{121}$$

valid for $d > 1$. The comparable quantity for independently sampled scaled Gaussian noise is

$$V_{\epsilon_1,\dots,\epsilon_K}[\widehat{\mathrm{tr}}(A)] = \frac{2d^2}{K(d+2)}\sigma^2(\lambda(A)) \tag{122}$$

i.e. $1/K$ times the $K = 1$ result. The orthogonalized noise strategy always has lower variance since the ratio between the variances is

$$\frac{d-K}{d-1} \leq 1 \tag{123}$$

which even reduces to zero when $K = d$. If $d$ is large, the difference is not great for small $K$, which aligns with the fact that randomly sampled directions in $\mathbb{R}^d$ are close to orthogonal for large $d$.

# E  EXPERIMENTAL DETAILS

## E.1  ROLE OF RECONSTRUCTION WEIGHT

### E.1.1  TOY DATA

To analyze the model behaviour depending on the reconstruction weight $\beta$, we train the same architecture on a simple sinusoid data set with $\beta$ varied between 0.01 and 100.

For the generation of data points, we draw $x$ positions from a 1D standard normal distribution and calculate the respective y positions by $y = \sin(\pi x/2)$. Then, isotropic Gaussian noise with $\sigma = 0.1$ is added. We train an autoencoder architecture built with four residual blocks and a 1D

---

[1]This argument is inspired by https://math.stackexchange.com/questions/1081345/finding-variance-of-the-sample-mean-of-a-random-sample-of-size-n-without-replace

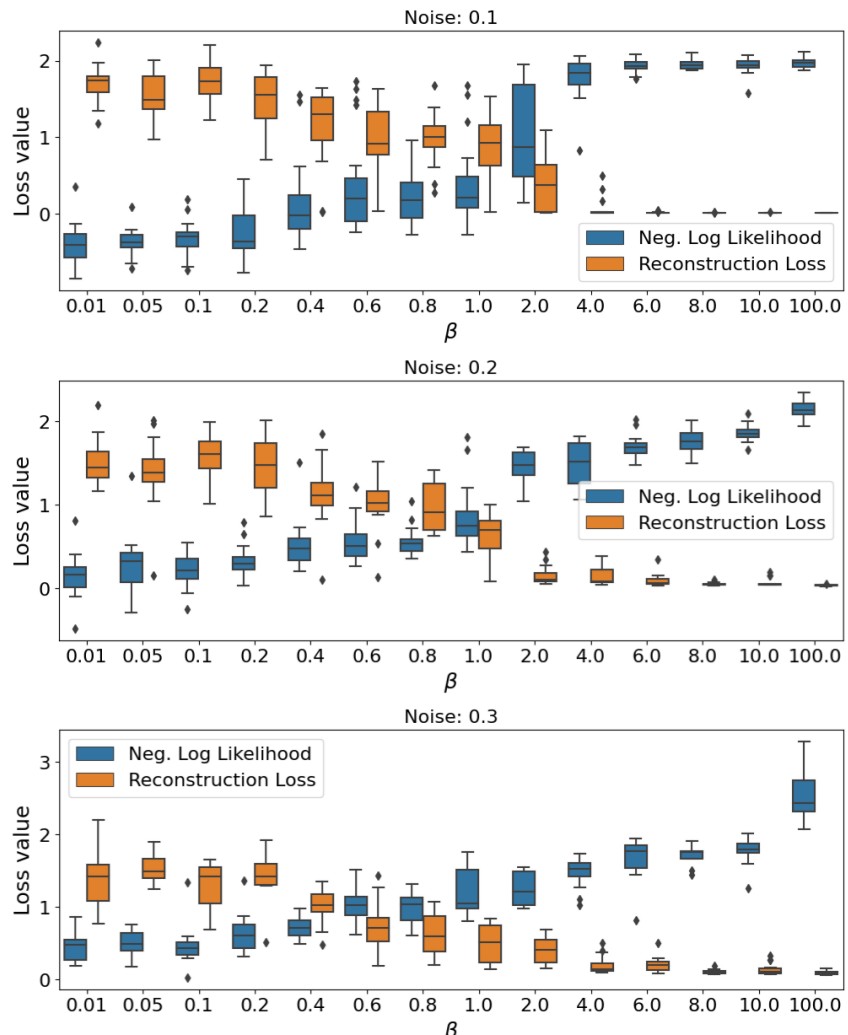

Figure 7: The position of the transition point depends on the data set. The plots show the tradeoff between reconstruction error and NLL with reconstruction weight $\beta$ (box plots summarize 20 runs per condition). The point at which $\beta$ becomes sufficiently large (transition point) shifts to lower values with increased Gaussian noise added to the data points.

latent space for 50 epochs with learning rate 0.001 until convergence. Each residual block is made up of a feedforward network with one hidden layer of width 256. For each value of $\beta$, 20 models are trained.

To visualize the dimension of the data that is captured by the model, we project samples from the data distribution to the (1D) latent space and color the data points using the respective latent code as color value. Figure 3 illustrates that low reconstruction weight $\beta$ values result in learning the dimension with the lowest entropy (noise) and higher values are required to learn the manifold that spans the sinusoid.

Additionally, we repeat the procedure with higher noise levels ($\sigma = 0.2, 0.3$). We observe that the point at which the model transitions from learning the noise to representing the manifold is not fixed, but depends on features of the data set such as the noise (fig. 7).

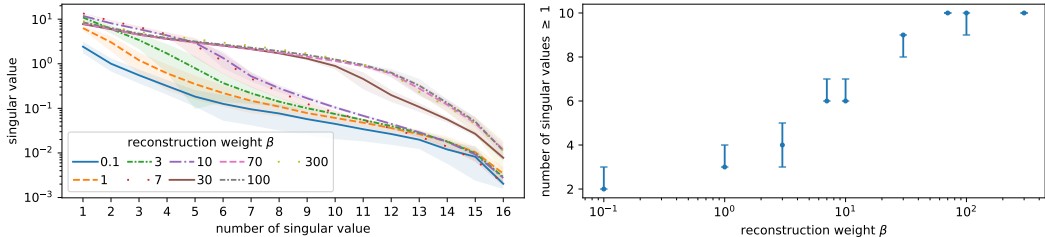

Figure 8: *(Left)* Singular value spectra for varying reconstruction weight. *(Right)* Number of singular values greater or equal to one as a function of reconstruction weight. The error bars show the span of the intersection of the shaded region with the line $y = 1$ in the left hand plot, rounded down to the nearest integer. For each trained model we generate 1024 samples per condition, compute the singular value spectra and average over all samples regardless of condition. The mean spectra and their standard deviation are evaluated across five trained models per reconstruction weight.



Figure 9: Conditional MNIST samples for varying reconstruction weight $\beta$. For each value of $\beta \in [0.1, 1, 10, 100]$ (from left to right), and each condition (rows) we generate ten samples (columns) at temperature $T = 1$.

### E.1.2 Conditional MNIST

To measure the structure of the conditional MNIST dataset learned by the generating model $g$, we compute the full decoder Jacobian matrix $J$ by calculating $d$ Jacobian-vector products (one per column of the Jacobian). We compute the singular values $\Sigma = \text{diag}(s_1, \ldots, s_d)$ of $J$, which – roughly speaking – indicate the stretching or shrinking of the latent manifold by $g$. Hence, the number of non-vanishing singular values suggest the dimension of the data manifold and the sum of the log singular values is equal to the change in entropy between the latent space and data space, where a higher entropy indicates that more of the data manifold is spanned by the decoder. We can see this from the formula

$$H(p_X) = H(p_Z) + E_{p_Z}\left[\frac{1}{2}\log\det(g'(z)^\top g'(z))\right] \tag{124}$$

with $H$ the differential entropy, and noting that $2\operatorname{tr}\log\Sigma = \log\det(g'(z)^\top g'(z))$.

We train multiple FIF models on conditional MNIST with reconstruction weights $\beta$ ranging from 0.1 to 100, and evaluate their singular value spectra. The models are trained for 400 epochs, which is sufficient for convergence. The architecture used is the same as in table 11, except that it has four times as many channels in each convolutional layer.

In fig. 8 it is clear that a higher reconstruction weight gives rise to a higher number of non-vanishing singular values. Hence, the reconstruction weight contributes towards learning structure of the true data manifold. This observed additional structure for higher reconstruction weights is reflected in an increasing diversity of samples (see fig. 9). Nevertheless, for high reconstruction weights we note the trade-off between sample diversity and properly learned latent distributions, which might result in out of distribution samples.

### E.2 Tabular data

We compare to the tabular data experiments in Caterini et al. (2021), using the same datasets and data splits, as well as the same latent space dimensions. We train models with roughly the same number

Table 5: Dataset-dependent hyperparameters and average total runtime for tabular data experiments. We compare our runtime against the published rectangular flow (Caterini et al., 2021) runtimes (RNFs-ML ($K = 1$) model), as well as rerunning their code on our hardware.

| Hyperparameter | POWER | GAS | HEPMASS | MINIBOONE |
|---|---|---|---|---|
| Latent dimension | 3 | 2 | 10 | 21 |
| Training epochs | 15 | 30 | 85 | 875 |
| Model | Training time (minutes) | | | |
| FIF (*ours*) | 38 | 39 | 41 | 49 |
| Rectangular flow (published) | 113 | 75 | 138 | 34 |
| Rectangular flow (our hardware) | 147 | 86 | 249 | 75 |
| Training time speedup (same hardware) | 3.9 $\times$ | 2.2 $\times$ | 6.1 $\times$ | 1.5 $\times$ |

Table 6: **Ablation study on the effect of each component of our proposed improvement** to rectangular flows (RF). By NLL estimator we denote how the loss in equation 3 is approximated. For this experiment we used our reimplementation of RF.

| Hyperparameters & Model | NLL estimator (on-/off-manifold) | POWER | GAS | HEPMASS | MINIBOONE |
|---|---|---|---|---|---|
| FIF & free-form net | off manifold (eq. 15) | **0.041 $\pm$ 0.007** | **0.281 $\pm$ 0.031** | **0.541 $\pm$ 0.034** | **0.598 $\pm$ 0.024** |
| FIF & free-form net | on manifold (eq. 10) | 19.54 $\pm$ 20.81 | 7.48 $\pm$ 5.40 | 29.03 $\pm$ 5.42 | 77.23 $\pm$ 16.55 |
| FIF & coupling flow | off manifold (eq. 15) | 0.11 $\pm$ 0.06 | 0.45 $\pm$ 0.09 | 1.30 $\pm$ 0.14 | 1.55 $\pm$ 0.04 |
| RF & coupling flow | off manifold (eq. 15) | 0.98 $\pm$ 0.69 | 6.16 $\pm$ 4.20 | 2.02 $\pm$ 0.74 | 1.80 $\pm$ 0.10 |
| FIF & coupling flow | on manifold (eq. 10) | 3.71 $\pm$ 2.19 | 0.40 $\pm$ 0.22 | 0.71 $\pm$ 0.05 | 3.13 $\pm$ 0.42 |
| RF & coupling flow | on manifold (eq. 10) | 0.33 $\pm$ 0.22 | 0.33 $\pm$ 0.17 | 0.82 $\pm$ 0.07 | 1.84 $\pm$ 0.11 |

of parameters. Our main architectural difference is that we use an unconstrained autoencoder rather than an injective flow. Our encoder consists of two parts: i) a feed-forward network with two hidden layers of dimension 256 and ReLU activations (no normalization layers), which maps from the input dimension to the latent dimension ii) a ResNet with two blocks, each with two hidden layers of dimension 256 and ReLU activations. The ResNet has input and output dimension equal to the latent space dimension. The decoder is the inverse: i) an identical ResNet to the encoder (but with separate parameters) followed by ii) a feed-forward network with two hidden layers of dimension 256 mapping from the latent space to the data space dimension.

We use a batch size of 512, add isotropic Gaussian noise with standard deviation 0.01, use $K = 1$ Hutchinson samples and a reconstruction weight $\beta = 10$ for all experiments. We use the Adam optimizer with the onecycle LR scheduler with LR of $10^{-4}$ (except for HEPMASS which has LR of $3 \times 10^{-4}$) and weight decay of $10^{-4}$. The number of epochs was chosen such that all experiments had approximately the same number of training iterations. We ran the model 5 times per dataset. The dataset-dependent parameters and average training times are given in table 5.

We compare our training times against the published rectangular flow training times for their RNFs-ML ($K = 1$) model, as well as rerunning their code on our hardware (a single RTX 2070 card). We find comparable FID-like scores on our rerun (except on GAS where we could not reproduce the score, see section 5), but our hardware is slower, with runs consistently taking at least 15% longer and more than twice as long on MINIBOONE. We find that our model runs in half the time or less of the rectangular flow on the same hardware, except for MINIBOONE (about 2/3 the time).

### E.3 COMPARISON TO EXISTING INJECTIVE FLOWS

We compare against Trumpets Kothari et al. (2021) and Denoising Normalizing Flows (DNF) Horvat & Pfister (2021), as they are the best-performing injective flows to our knowledge, and report performance on CelebA in table 3. Note that Trumpets default to $d = 192$, DNF to $d = 512$, whereas we are able to reduce the bottleneck dimension to $d = 64$ (consistent with the Pythae benchmark in appendix E.4).

Table 7: Reconstruction losses of FIF with a free-form architecture on the POWER, GAS HEP-MASS and MINIBOONE datasets. The reconstruction error is always much higher for on-manifold training compared to off-manifold, demonstrating the instability caused by on-manifold NLL evaluation in free-form networks. Note: the large standard deviations in on-manifold runs are typically the result of a single large outlier. We remove the largest outlier where applicable ("On Manifold (outliers removed)" row).

|  | POWER | GAS | HEPMASS | MINIBOONE |
|---|---|---|---|---|
| On manifold | $237 \pm 498$ | $5835 \pm 13006$ | $119 \pm 34$ | $300 \pm 160$ |
| On manifold (outliers removed) | $14 \pm 27$ | $18 \pm 31$ | $119 \pm 34$ | $229 \pm 23$ |
| Off manifold | $0.072 \pm 0.002$ | $0.188 \pm 0.012$ | $2.569 \pm 0.098$ | $1.077 \pm 0.011$ |

Both models differ in the recommended wall clock time, and we therefore fix the wall clock time available to each model to five hours on a single NVIDIA A40. Trumpets train the manifold and the distribution on it in two sequential steps. To accommodate both steps in the reduced training time, we vary the fraction of the five hours spent in training manifold and distribution and report the best FID among the variants tried. We vary number of manifold epochs as $n_{\mathrm{manifold}} = 2, 5, 10$, with 10 performing best.

Our free-form injective flows (FIF) are not restricted in their architecture, and we choose an off-the-shelf convolutional autoencoder, followed by a total of four fully-connected ResNet blocks, see table 11. The fully-connected blocks are important, as can be seen when comparing to the architecture used in the Pythae benchmark (see appendix E.4). We note that the Pythae benchmark could benefit from a modified architecture, but leave this modification open for future work.

For Trumpets and DNF, we point to the training details provided by the respective authors. For FIF, we choose these training hyperparameters: We train with the Adam optimizer with a LR of $10^{-3}$ and a weight decay of $10^{-4}$, linearly increase $\beta$ from 20 at initialization to 40 at the end of training, a single Hutchinson sample $K = 1$ and a student-t distribution on the latent space. We set the batch size to 256.

We conclude that the Pythae benchmark could benefit from an optimized architecture, as this change probably also improves the other methods. From the data at hand, we further conclude that the full potential of FIF has not yet been exploited, and that easy gains can be made by improving the architecture and other hyperparameters.

### E.4 PYTHAE BENCHMARK ON GENERATIVE AUTOENCODERS

We compare our method to existing generative autoencoder paradigms using the benchmark from Chadebec et al. (2022). We use the provided open-source pipeline and follow the training setup described by the authors. For MNIST and CIFAR10 this means training for 100 epochs with the Adam optimizer at a starting LR of $10^{-4}$, reserving the last 10k images of the training sets as validation sets. CelebA trains for 50 epochs with a starting LR of $10^{-3}$. All experiments are performed with a batch size of 100 and LR is reduced by half when the loss plateaus for 10 epochs. In accordance with the original benchmark we pick 10 sets of hyper-parameters, compute their validation FID (see fig. 10) and use the model which achieves the best FID on the validation set as the final model. In table 8 we report the FID and IS of this model on the test set. To complement the metrics, we show samples from all models in fig. 11, demonstrating convincing quality. We exclude the VAEGAN from the FID comparison, as the model trains more than double the time required for FIF and goes beyond fitting a transformation of the training data to a standard normal distribution.

As described in section 5, we use the architectures from Chadebec et al. (2022) for the benchmark, which we replicate in tables 9 and 10.

### E.5 COMPUTE AND DEPENDENCIES

We used approximately 150 GPU hours for computing the Pythae benchmark, and an additional 800 GPU hours for model exploration and testing. The majority of the experiments were performed on an internal cluster of A100s. The majority of compute time was spent on image datasets.

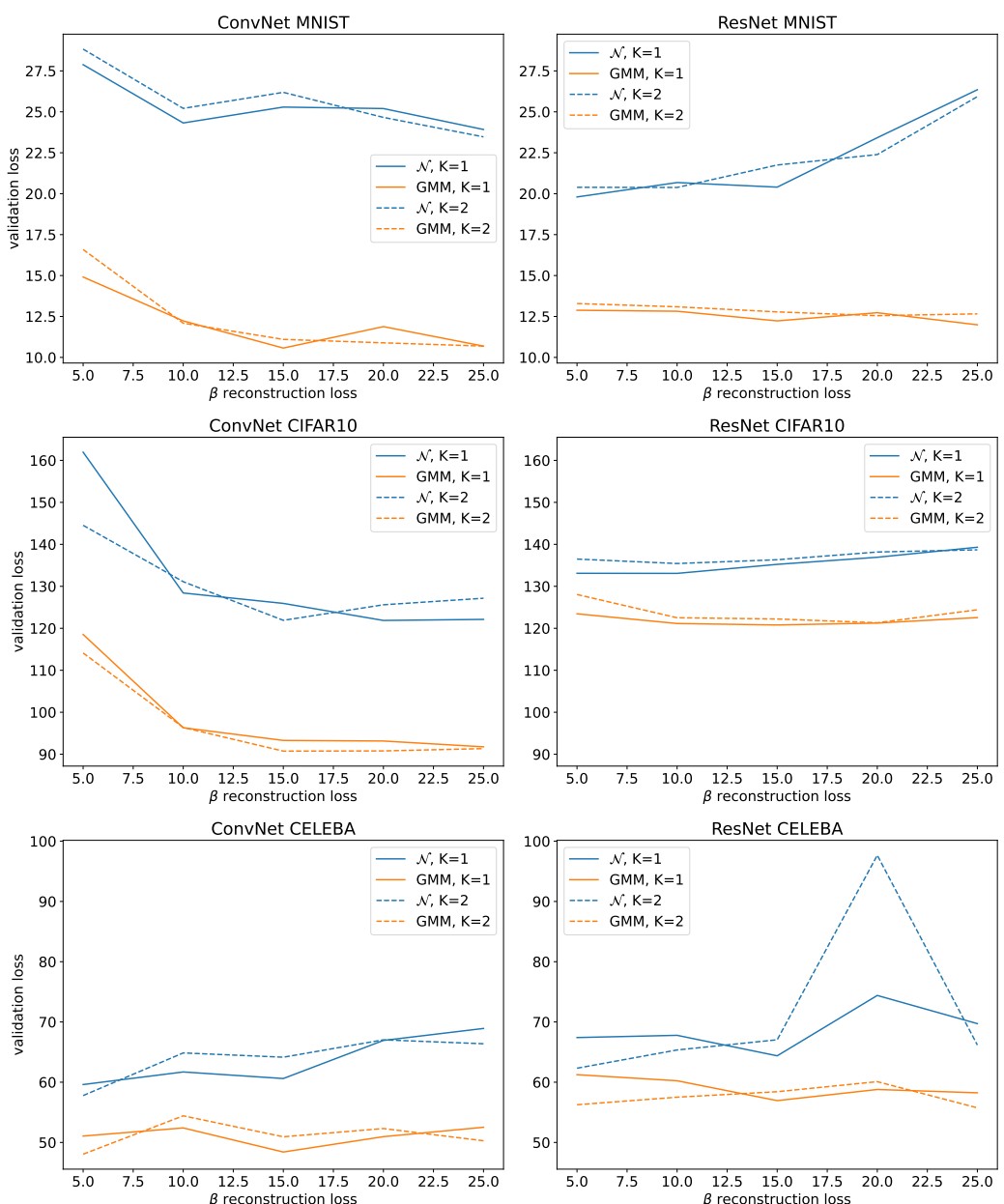

Figure 10: Validation FID of our 10 benchmark model setups on the different datasets and architectures used by Pythae. We report results for reconstruction weights $\beta = 5, 10, 15, 20, 25$ and number of Hutchinson samples $K = 1$ in the solid lines and $K = 2$ in the dashed lines. We show the performance of standard normal sampling ($\mathcal{N}$) and a Gaussian mixture model (GMM). We see that GMM sampling always improves performance. In contrast, the reconstruction weight and number of Hutchinson samples have no noticeable effect on performance, except that increasing $\beta$ improves performance on ConvNet MNIST and ConvNet CIFAR10, and decreases performance on ResNet MNIST (normal sampling only).

We build our code upon the following python libraries: PyTorch (Paszke et al., 2019), PyTorch Lightning (Falcon & The PyTorch Lightning team, 2019), Tensorflow (Abadi et al., 2015) for FID score evaluation, Numpy (Harris et al., 2020), Matplotlib (Hunter, 2007) for plotting and Pandas (McKinney, 2010; The pandas development team, 2020) for data evaluation.

Table 8: Table taken from Chadebec et al. (2022) with our results added at the top. We report Inception Score (IS) and Fréchet Inception Distance (FID) computed with 10k samples on the test set. The best model per dataset and sampler is highlighted in **bold**, the second best is underlined.

| | | ConvNet | | | | | | ResNet | | | | | |
| | | MNIST | | CIFAR10 | | CELEBA | | MNIST | | CIFAR10 | | CELEBA | |
| Model | Sampler | FID ↓ | IS ↑ | FID | IS | FID | IS ↑ | FID ↓ | IS ↑ | FID | IS | FID | IS |
|---|---|---|---|---|---|---|---|---|---|---|---|---|---|
| FIF (ours) | $\mathcal{N}$ | 23.8 | 2.2 | 121.0 | 3.0 | 56.9 | 2.1 | 19.5 | 2.1 | 132.6 | 2.9 | **62.3** | 1.7 |
| | GMM | 11.0 | 2.2 | 90.6 | 4.0 | **47.3** | 1.9 | 11.7 | 2.1 | 119.2 | 3.4 | **55.0** | 1.8 |
| VAE | $\mathcal{N}$ | 28.5 | 2.1 | 241.0 | 2.2 | 54.8 | 1.9 | 31.3 | 2.0 | 181.7 | 2.5 | 66.6 | 1.6 |
| | GMM | 26.9 | 2.1 | 235.9 | 2.3 | 52.4 | 1.9 | 32.3 | 2.1 | 179.7 | 2.5 | 63.0 | 1.7 |
| VAMP | VAMP | 64.2 | 2.0 | 329.0 | 1.5 | 56.0 | 1.9 | 34.5 | 2.1 | 181.9 | 2.5 | 67.2 | 1.6 |
| IWAE | $\mathcal{N}$ | 29.0 | 2.1 | 245.3 | 2.1 | 55.7 | 1.9 | 32.4 | 2.0 | 191.2 | 2.4 | 67.6 | 1.6 |
| | GMM | 28.4 | 2.1 | 241.2 | 2.1 | 52.7 | 1.9 | 34.4 | 2.1 | 188.8 | 2.4 | 64.1 | 1.7 |
| VAE-lin NF | $\mathcal{N}$ | 29.3 | 2.1 | 240.3 | 2.1 | 56.5 | 1.9 | 32.5 | 2.0 | 185.5 | 2.4 | 67.1 | 1.6 |
| | GMM | 28.4 | 2.1 | 237.0 | 2.2 | 53.3 | 1.9 | 33.1 | 2.1 | 183.1 | 2.5 | 62.8 | 1.7 |
| VAE-IAF | $\mathcal{N}$ | 27.5 | 2.1 | 236.0 | 2.2 | 55.4 | 1.9 | 30.6 | 2.0 | 183.6 | 2.5 | 66.2 | 1.6 |
| | GMM | 27.0 | 2.1 | 235.4 | 2.2 | 53.6 | 1.9 | 32.2 | 2.1 | 180.8 | 2.5 | 62.7 | 1.7 |
| $\beta$-VAE | $\mathcal{N}$ | 21.4 | 2.1 | **115.4** | 3.6 | 56.1 | 1.9 | **19.1** | 2.0 | **124.9** | 3.4 | 65.9 | 1.6 |
| | GMM | 9.2 | 2.2 | 92.2 | 3.9 | 51.7 | 1.9 | 11.4 | 2.1 | 112.6 | 3.6 | 59.3 | 1.7 |
| Dis $\beta$-VAE | $\mathcal{N}(0, 1)$ | 96.5 | 2.3 | 219.4 | 3.6 | 130.8 | 1.6 | 109.4 | 2.7 | 209.8 | 3.2 | 110.6 | 1.5 |
| | GMM | 192.7 | 2.3 | 300.8 | 1.8 | 94.0 | 1.5 | 98.7 | 2.1 | 202.3 | 2.6 | 83.2 | 1.6 |
| | 2-s sampler | 236.1 | 1.5 | 371.2 | 1.0 | 167.9 | 1.0 | 250.8 | 1.1 | 349.0 | 1.2 | 161.0 | 1.2 |
| | MAF sampler | 191.8 | 2.2 | 300.6 | 1.8 | 94.3 | 1.5 | 98.3 | 2.1 | 200.6 | 2.6 | 82.6 | 1.7 |
| $\beta$-TC VAE | $\mathcal{N}$ | 21.3 | 2.1 | 116.6 | 2.8 | 55.7 | 1.8 | 20.7 | 2.0 | 125.8 | 3.4 | 65.9 | 1.6 |
| | GMM | 11.6 | 2.2 | 89.3 | 4.1 | 51.8 | 1.9 | 13.3 | 2.1 | **106.5** | 3.7 | 59.3 | 1.7 |
| FactorVAE | $\mathcal{N}$ | 27.0 | 2.1 | 236.5 | 2.2 | 53.8 | 1.9 | 31.0 | 2.0 | 185.4 | 2.5 | 66.4 | 1.7 |
| | GMM | 26.9 | 2.1 | 234.0 | 2.2 | 52.4 | 2.0 | 32.7 | 2.1 | 184.4 | 2.5 | 63.3 | 1.7 |
| InfoVAE - RBF | $\mathcal{N}$ | 27.5 | 2.1 | 235.2 | 2.1 | 55.5 | 1.9 | 31.1 | 2.0 | 182.8 | 2.5 | 66.5 | 1.6 |
| | GMM | 26.7 | 2.1 | 230.4 | 2.2 | 52.7 | 1.9 | 32.3 | 2.1 | 179.5 | 2.5 | 62.8 | 1.7 |
| InfoVAE - IMQ | $\mathcal{N}$ | 28.3 | 2.1 | 233.8 | 2.2 | 56.7 | 1.9 | 31.0 | 2.0 | 182.4 | 2.5 | 66.4 | 1.6 |
| | GMM | 27.7 | 2.1 | 231.9 | 2.2 | 53.7 | 1.9 | 32.8 | 2.1 | 180.7 | 2.6 | 62.3 | 1.7 |
| AAE | $\mathcal{N}$ | **16.8** | 2.2 | 139.9 | 2.6 | 59.9 | 1.8 | **19.1** | 2.1 | 164.9 | 2.4 | 64.8 | 1.7 |
| | GMM | 9.3 | 2.2 | 92.1 | 3.8 | 53.9 | 2.0 | 11.1 | 2.1 | 118.5 | 3.5 | 58.7 | 1.8 |
| MSSSIM-VAE | $\mathcal{N}$ | 26.7 | 2.2 | 279.9 | 1.7 | 124.3 | 1.3 | 28.0 | 2.1 | 254.2 | 1.7 | 119.0 | 1.3 |
| | GMM | 27.2 | 2.2 | 279.7 | 1.7 | 124.3 | 1.3 | 28.8 | 2.1 | 253.1 | 1.7 | 119.2 | 1.3 |
| VAEGAN (not compared) | $\mathcal{N}$ | *8.7* | *2.2* | *199.5* | *2.2* | *39.7* | *1.9* | *12.8* | *2.2* | *198.7* | *2.2* | *122.8* | *2.0* |
| | GMM | *6.3* | *2.2* | *197.5* | *2.1* | *35.6* | *1.8* | *6.5* | *2.2* | *188.2* | *2.6* | *84.3* | *1.7* |
| AE | $\mathcal{N}$ | 26.7 | 2.1 | 201.3 | 2.1 | 327.7 | 1.0 | 221.8 | 1.3 | 210.1 | 2.1 | 275.0 | 2.9 |
| | GMM | 9.3 | 2.2 | 97.3 | 3.6 | 55.4 | 2.0 | 11.0 | 2.1 | 120.7 | 3.4 | 57.4 | 1.8 |
| WAE - RBF | $\mathcal{N}$ | 21.2 | 2.2 | 175.1 | 2.0 | 332.6 | 1.0 | 21.2 | 2.1 | 170.2 | 2.3 | 69.4 | 1.6 |
| | GMM | 9.2 | 2.2 | 97.1 | 3.6 | 55.0 | 2.0 | 11.2 | 2.1 | 120.3 | 3.4 | 58.3 | 1.7 |
| WAE - IMQ | $\mathcal{N}$ | 18.9 | 2.2 | 164.4 | 2.2 | 64.6 | 1.7 | 20.3 | 2.1 | 150.7 | 2.5 | 67.1 | 1.6 |
| | GMM | **8.6** | 2.2 | 96.5 | 3.6 | 51.7 | 2.0 | 11.2 | 2.1 | 119.0 | 3.5 | 57.7 | 1.8 |
| VQVAE | $\mathcal{N}$ | 28.2 | 2.0 | 152.2 | 2.0 | 306.9 | 1.0 | 170.7 | 1.6 | 195.7 | 1.9 | 140.3 | 2.2 |
| | GMM | 9.1 | 2.2 | 95.2 | 3.7 | 51.6 | 2.0 | **10.7** | 2.1 | 120.1 | 3.4 | 57.9 | 1.8 |
| RAE - L2 | $\mathcal{N}$ | 25.0 | 2.0 | 156.1 | 2.6 | 86.1 | 2.8 | 63.3 | 2.2 | 170.9 | 2.2 | 168.7 | 3.1 |
| | GMM | 9.1 | 2.2 | **85.3** | 3.9 | 55.2 | 1.9 | 11.5 | 2.1 | 122.5 | 3.4 | 58.3 | 1.8 |
| RAE - GP | $\mathcal{N}$ | 27.1 | 2.1 | 196.8 | 2.1 | 86.1 | 2.4 | 61.5 | 2.2 | 229.1 | 2.0 | 201.9 | 3.1 |
| | GMM | 9.7 | 2.2 | 96.3 | 3.7 | 52.5 | 1.9 | 11.4 | 2.1 | 123.3 | 3.4 | 59.0 | 1.8 |

Table 9: **ConvNet**, neural network architecture used for the convolutional networks, adapted from Chadebec et al. (2022).

| | MNIST | CIFAR10 | CELEBA |
|---|---|---|---|
| Encoder | (1, 28, 28) | (3, 32, 32) | (3, 64, 64) |
| Layer 1 | Conv(128, 4, 2), BN, ReLU | Conv(128, 4, 2), BN, ReLU | Conv(128, 4, 2), BN, ReLU |
| Layer 2 | Conv(256, 4, 2), BN, ReLU | Conv(256, 4, 2), BN, ReLU | Conv(256, 4, 2), BN, ReLU |
| Layer 3 | Conv(512, 4, 2), BN, ReLU | Conv(512, 4, 2), BN, ReLU | Conv(512, 4, 2), BN, ReLU |
| Layer 4 | Conv(1024, 4, 2), BN, ReLU | Conv(1024, 4, 2), BN, ReLU | Conv(1024, 4, 2), BN, ReLU |
| Layer 5 | Linear(1024, latent_dim)* | Linear(4096, latent_dim)* | Linear(16384, latent_dim)* |
| Decoder | | | |
| Layer 1 | Linear(latent_dim, 16384) | Linear(latent_dim, 65536) | Linear(latent_dim, 65536) |
| Layer 2 | ConvT(512, 3, 2), BN, ReLU | ConvT(512, 4, 2), BN, ReLU | ConvT(512, 5, 2), BN, ReLU |
| Layer 3 | ConvT(256, 3, 2), BN, ReLU | ConvT(256, 4, 2), BN, ReLU | ConvT(256, 5, 2), BN, ReLU |
| Layer 4 | Conv(1, 3, 2), Sigmoid | Conv(3, 4, 1), Sigmoid | ConvT(128, 5, 2), BN, ReLU |
| Layer 5 | - | - | ConvT(3, 5, 1), Sigmoid |
| #Parameters | 17.2M | 39.4M | 33.5M |

Table 10: **ResNet**, neural network architecture used for the residual networks, adapted from Chadebec et al. (2022).

|  | MNIST | CIFAR10 | CELEBA |
|---|---|---|---|
| Encoder | (1, 28, 28) | (3, 32, 32) | (3, 64, 64) |
| Layer 1 | Conv(64, 4, 2) | Conv(64, 4, 2) | Conv(64, 4, 2) |
| Layer 2 | Conv(128, 4, 2) | Conv(128, 4, 2) | Conv(128, 4, 2) |
| Layer 3 | Conv(128, 3, 2) | Conv(128, 3, 1) | Conv(128, 3, 2) |
| Layer 4 | ResBlock* | ResBlock* | Conv(128, 3, 2) |
| Layer 5 | ResBlock* | ResBlock* | ResBlock* |
| Layer 6 | Linear(2048, latent_dim)* | Linear(8192, latent_dim)* | ResBlock* |
| Layer 7 | - | - | Linear(2048, latent_dim)* |
| Decoder |  |  |  |
| Layer 1 | Linear(latent_dim, 2048) | Linear(latent_dim, 8192) | Linear(latent_dim, 2048) |
| Layer 2 | ConvT(128, 3, 2) | ResBlock* | ConvT(128, 3, 2) |
| Layer 3 | ResBlock* | ResBlock* | ResBlock* |
| Layer 4 | ResBlock*, ReLU | ConvT(64, 4, 2) | ResBlock* |
| Layer 5 | ConvT(64, 3, 2), ReLU | ConvT(3, 4, 2), Sigmoid | ConvT(128, 5, 2), Sigmoid |
| Layer 6 | ConvT(1, 3, 2), Sigmoid | - | ConvT(64, 5, 2), Sigmoid |
| Layer 6 | - | - | ConvT(3, 4, 2), Sigmoid |
| #Parameters | 0.73M | 4.8M | 1.6M |

*The ResBlocks are composed of one Conv(32, 3, 1) followed by Conv(128, 1, 1) with ReLU.

Table 11: **FIF-ConvNet**, neural network architecture used for comparison to Trumpet and Denoising Normalizing Flow.

|  | MNIST | CELEBA |
|---|---|---|
| Encoder | (1, 28, 28) | (3, 64, 64) |
| Layer 1 | Conv(32, 4, 2), BN, ReLU | Conv(128, 4, 2), BN, ReLU |
| Layer 2 | Conv(64, 4, 2), BN, ReLU | Conv(256, 4, 2), BN, ReLU |
| Layer 3 | Conv(128, 4, 2), BN, ReLU | Conv(512, 4, 2), BN, ReLU |
| Layer 4 | Conv(256, 4, 2), BN, ReLU | Conv(*1024*, 4, 2), BN, ReLU |
| Layer 5 | Linear(256, latent_dim)* | Linear(16384, latent_dim)* |
| Layer 6-9 | 4xResBlock(512) | 4xResBlock(256) |
| Decoder |  |  |
| *Layer 1-4* | *4xResBlock(512)* | *4xResBlock(256)* |
| Layer 5 | Linear(latent_dim, *4096*) | Linear(latent_dim, 65536) |
| Layer 6 | ConvT(256, 3, 2), BN, ReLU | ConvT(512, 5, 2), BN, ReLU |
| Layer 7 | ConvT(128, 3, 2), BN, ReLU | ConvT(256, 5, 2), BN, ReLU |
| Layer 8 | ConvT(64, 3, 2), BN, ReLU | ConvT(128, 5, 2), BN, ReLU |
| Layer 9 | Conv(1, 3, 2), Sigmoid | ConvT(3, 5, 1), Sigmoid |
| #Parameters | 3.3M | 34.3M |

*The ResBlocks(inner_dim) are composed of Linear(latent_dim, inner_dim), SiLU, Linear(inner_dim, inner_dim), SiLU, Linear(inner_dim, latent_dim) with a skip connection.

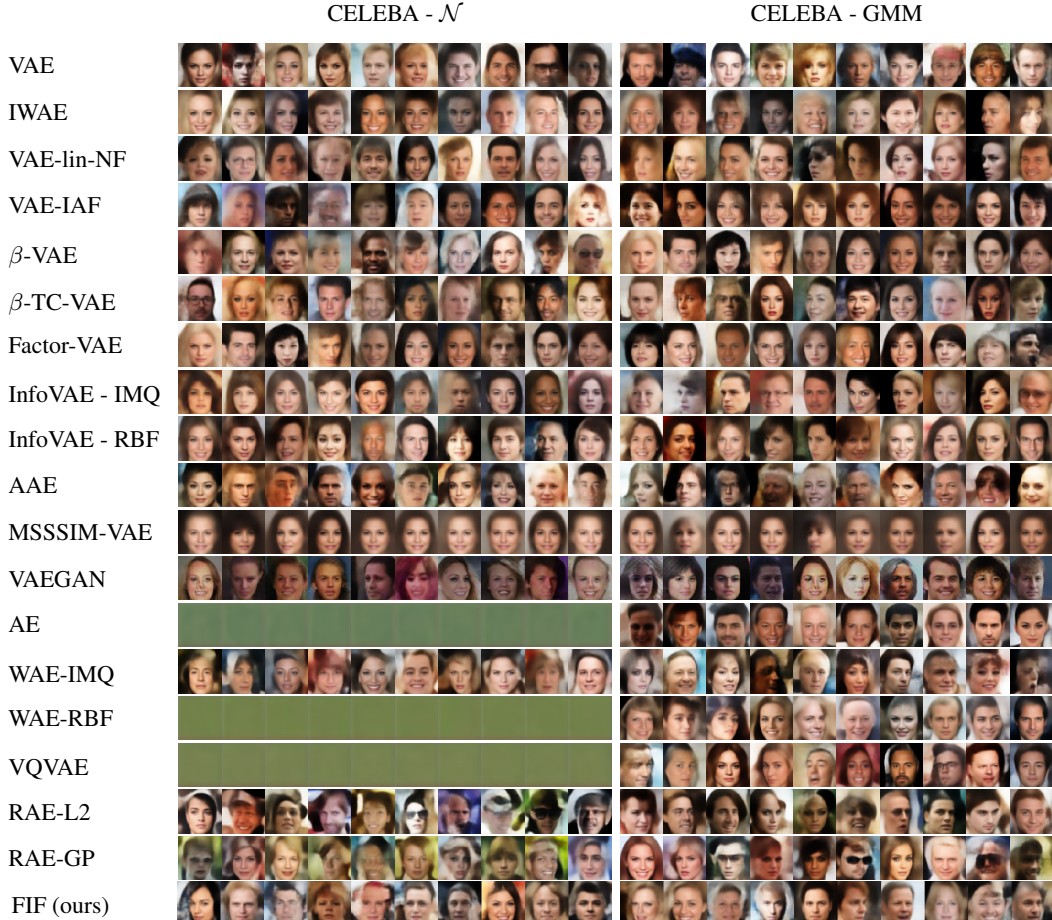

Figure 11: Uncurated samples from the CelebA ConvNet experiments in the PythAE benchmark. Our model is shown at the bottom, samples from the other models have been taken from Chadebec et al. (2022).

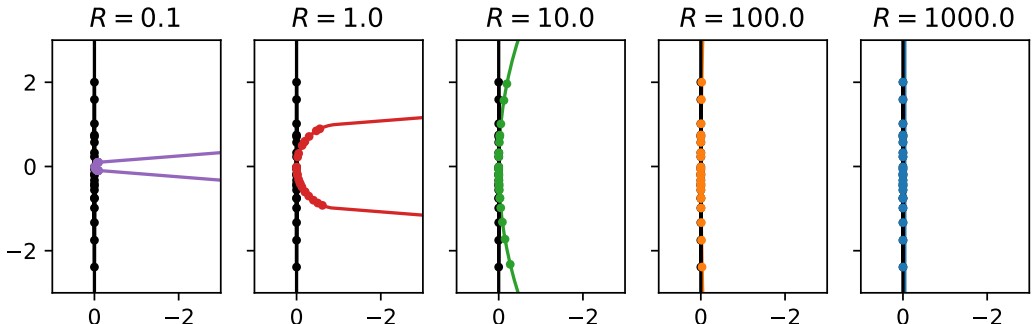

Figure 12: Possible learnt manifolds of varying curvature $1/R$ for data supported on a subspace, where $d = 1$ and $D = 2$.

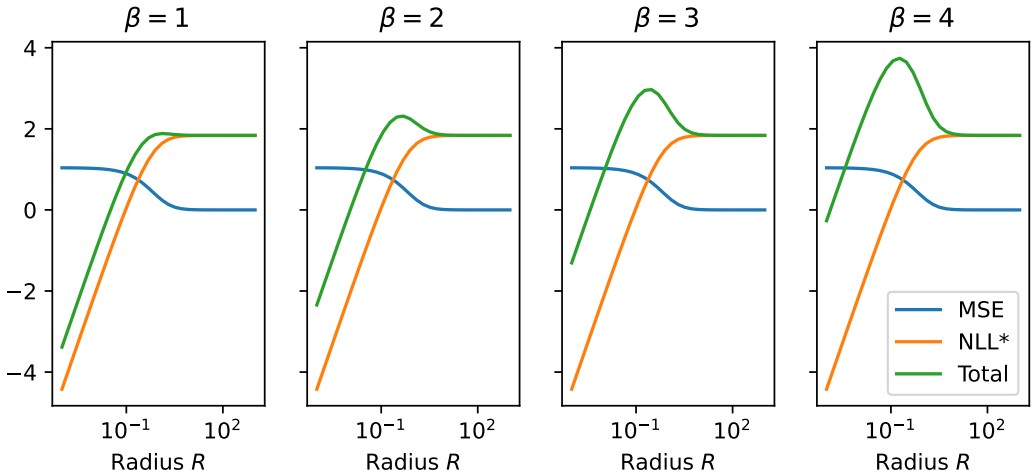

Figure 13: Weighting the reconstruction loss higher does not lead to stable training. The plots show different reconstruction weights $\beta$. In all settings, highly curved manifolds (i.e. low radius) achieve the lowest loss.

## F    DETAILS ON PATHOLOGY INDUCED BY CURVATURE

As described in section 4.2, gradients from the on-manifold loss in eq. (10) cause the learned manifold to increase curvature. This is visualized in the main text in fig. 2, where the left plot shows that this loss leads to ever-increasing curvature. The reason is that the entropy of data projected to a curved manifold is smaller than the entropy of data projected to a flat manifold.

Here, we provide intuition for why this happens for synthetic data where $d$ is known and the data could in principle be perfectly reconstructed. Figure 12 depicts projections of the data to possible model manifolds of varying curvature $\kappa$. We parameterize the curvature by varying the radius $R = 1/\kappa$. One can observe that for with increasing curvature (i.e. decreasing radius), the data is projected to an increasingly small region. Correspondingly, the entropy $H(\hat{p}_{\text{data}}(\hat{x}))$ of the projected data becomes arbitrarily negative (just like a Gaussian with low standard deviation has arbitrarily negative entropy), lowering the achievable negative log-likelihood.

Adding reconstruction loss alone does not fix this pathology, which we illustrate in fig. 13: The reconstruction loss saturates for small radii, but the best achievable negative log-likelihood (i.e. the entropy of the data) continues to decrease with the radius. Thus, even when increasing $\beta$, the minimal possible value of the total loss is still achieved by a spuriously curved manifold.

