# OpenReview forum: "Lifting Architectural Constraints of Injective Flows"
_ICLR.cc/2024/Conference — ICLR 2024 poster_

### Official Review · Reviewer_RfcP · 2023-10-19

**Soundness:** 3 good
**Presentation:** 3 good
**Contribution:** 2 fair
**Rating:** 5
**Confidence:** 4

**Summary:**

- The authors propose a new kind of injective flow (a normalizing flow with lower-dimensional latent space than the data space).
- Unlike some previous attempts, they use unconstrained encoders and decoders, and introduce a new estimator for the gradient of the change-of-variables term in the log likelihood.
- The authors also discuss a training issue for injective flows that was previously pointed out by Brehmer & Cranmer (2020).
- They demonstrate the method by training generative models on tabular data and CelebA, reporting metrics of generative quality like FID.

**Strengths:**

1. The main idea (the new estimator of the gradient) is wonderfully simple, sensible, and efficient.
2. The paper is clearly structured and well-written.

**Weaknesses:**

1. Injective flows are academically interesting, but do not have a very clear use case, especially if they do not have a tractable density (see questions below).
2. The discussion of joint manifold and likelihood training is not novel (see Brehmer & Cranmer, 2020), which the authors are open about. The proposed solution leaves questions open (see below).
3. I am not yet convinced by the experimental evaluation. Given the quality of the samples in Figure 1, I am surprised by the claim that the method outperforms various VAE methods (see questions below).
4. Overall, the paper's contributions are quite thin.

**Questions:**

1. What's the main use case for this injective flow? In what situations do you expect benefits from the manifold structure of this generative model compared to, say, diffusion models or VAEs?
2. Is the density (not its gradient) of the model tractable? That would extend use cases substantially.
3. I don't understand the "fix" of the pathological behaviour pointed out in Sec. 4.2. Could you expand the discussion of why it would work? Is it guaranteed to work? Consider the toy problem that Brehmer & Cranmer (2020) use to illustrate the same problem (Fig 4 in the arXiv version, 2003.13913). Here the encoder $f(x)$ is linear, thus $f'(\hat{x}) = f'(x)$ , and the "fix" does not change anything.
4. Do you have an explanation for why the problem discussed in Sec. 4.2 does not affect the experiments?
5. In the experimental evaluation, are the models converged? How do the results change if the models (in particular the baselines) are trained for longer? I find it hard to believe that none of the VAE methods are able to produce higher-quality samples than what we see in Fig. 1.

---

> ### Author Response · Authors · 2023-11-15
>
> Thank you for your constructive feedback. Let us answer to your questions, and we hope that they also soften the mentioned weaknesses:
>
> 1. FIF are fast, perform well and are simple to set up:
>     1. FIF vs diffusion model: Diffusion models sample slow, which makes them expensive in inference.
>     2. FIF vs VAE: VAEs are limited by (i) their variational family, and (ii) by learning a distribution in the embedding space, which is limiting if the data actually lies on a manifold (cf. manifold overfitting). While (i) can be addressed by more expressive likelihoods/posteriors, this comes with additional hyperparameters. Our model only has the architecture and beta as hyperparameters.
>     3. FIF vs GAN: FIF is trained via maximum likelihood, so it guarantees against mode collapse.
>     4. FIF vs latent flow/diffusion/…, e.g. Stable Diffusion: FIF trains manifold and distribution jointly, while post-hoc learning the latent distribution is a two-step procedure.
> 2. The density is tractable by computing the Jacobian of the transformation using as many autograd calls as there are latent dimensions. While this may not be computationally feasible for very large latent dimensions, we find that for all problems considered it is still fine. However, there are unbiased estimators for the change of variables which require less autograd evaluations [Chen et al. 2019: Residual Flows for Invertible Generative Modeling].
> 3. We do not simply repeat the problem mentioned in Brehmer & Cranmer, this seems to be a misunderstanding. Let us give a different perspective on the well-behaved loss in section 4.2. We need to address *two* pathologies, each with a different solution. The first was described by Brehmer & Cranmer and appears already for a linear model, and the second one is new and only applies to nonlinear models. You correctly noted that our fix to the second problem does not help against the first problem. So let us quickly sum up the two pathologies and the fix to each.
>     1. **Degenerate subspaces** (identified by Brehmer & Cranmer): If the data is only supported on a low-dimensional subspace, then there is a projection of the data with $-\infty$ log-likelihood. This possible projection can be eliminated by adding a tiny bit of noise to the data. In Figure 3, we demonstrate that a high enough reconstruction weight $\beta$ is sufficient to avoid learning degenerate projections. We also present an analytic result on the linear case in Appendix C.
>     2. **Strong curvature** (new): If the learnt manifold curves strongly, then data on the outside of the curve is concentrated by projecting to the manifold (see Figure 2). This decreases the entropy of $p(\hat x)$, lowering the lower bound of the manifold log-likelihood, causing divergence. We fix this by computing the encoder Jacobian off-manifold, which punishes spurious curvature (see discussion in appendix B.2). This is also cheaper because it avoids one forward pass through the model.
> 4. See Q3: We provide a fix for both pathologies.
> 5. We decided to use the Pythae benchmark because it allows a systematic comparison to a large collection of generative autoencoders. However, it puts a compute limit on training by fixing the architecture and the number of training epochs. The purpose of the benchmark is to fairly compare methods, not to achieve SOTA results. This explains the sample quality, see Figure 11 in their paper for samples from competitor models: https://arxiv.org/pdf/2206.08309.pdf
>
> We hope that this addresses your concerns. We are happy to provide further information.

---

> > ### Comment · Reviewer_RfcP · 2023-11-20
> >
> > Thank you for the detailed responses.
> >
> > > 2. The density is tractable by computing the Jacobian of the transformation using as many autograd calls as there are latent dimensions.
> >
> > Do you here need to assumesthat the model actually learns to be injective? Can you verify if this is the case for a given trained model, without knowledge of the data manifold?
> >
> > > 3. We do not simply repeat the problem mentioned in Brehmer & Cranmer, this seems to be a misunderstanding. [...] We need to address two pathologies, each with a different solution.
> >
> > Ah, thanks for clearing this up, I indeed misunderstood the discussion here. So the additional problem you describe only occurs when the data does does not actually populate a lower-dimensional manifold, but rather extends off-manifold into the ambient space, and an injective flow will never be able to achieve perfect reconstruction – correct? Then I agree, this is actually a new pathology, and the solution offered by Brehmer & Cranmer does not apply.
> >
> > How relevant is this problem, though? I was under the impression that the manifold hypothesis / zero reconstruction error is quite central in motivating this work. In your experiments, what reconstruction errors did you achieve?
> >
> > I also still think that both pathologies are two instances of the same underlying root cause (optimizing likelihoods after a learnable projection).
> >
> > Thanks again for your response. While I still have some doubts about the empirical performance and motivation of the method, I now see the contribution of this paper more clearly and will update my score to reflect this.

---

> > > ### Author Response · Authors · 2023-11-21
> > >
> > > Thank you for your continued feedback and questions!
> > >
> > > > Do you here need to assume that the model actually learns to be injective? Can you verify if this is the case for a given trained model, without knowledge of the data manifold?
> > >
> > > Yes, we can measure how well injectivity has been learned by reconstructing the reconstructions once more – under perfect convergence, they should remain identical.
> > > Note that minimal reconstruction does not mean that the reconstruction loss is zero, but it means that the reconstruction is as good as it can get with the available number of latent dimensions.
> > >
> > > > So the additional problem you describe only occurs when the data does does not actually populate a lower-dimensional manifold, but rather extends off-manifold into the ambient space, and an injective flow will never be able to achieve perfect reconstruction – correct?
> > >
> > > The additional problem occurs regardless of whether the data is supported on a subspace or extends off-manifold. Consider figure 2. The pathological behavior illustrated on the left could still happen if the data distribution were a Gaussian with zero variance in the x direction, since the concentration effect due to curvature would still overwhelm the reconstruction error.
> > >
> > > > How relevant is this problem, though? I was under the impression that the manifold hypothesis / zero reconstruction error is quite central in motivating this work. In your experiments, what reconstruction errors did you achieve?
> > >
> > > In our experiments, this problem seems to be important: If we do not apply our fix, training is unstable and we find substantially worse results. Thanks to the idea of reviewer SnSS, we performed an ablation during this rebuttal that compared the performance with and without the fix. The table below takes the numbers from our Table 1 and adds an additional row which shows the performance on the tabular data of a free-form architecture if the loss is evaluated on-manifold (i.e. without our fix):
> > >
> > > | Experiment                               | power         | gas         | hepmass      | miniboone     |
> > > |------------------------------------------|---------------|-------------|--------------|---------------|
> > > | RF: injective flow via RF loss with CG   | 0.08 ± 0.02   | 0.11 ± 0.02 | 0.78 ± 0.19  | 1.00 ± 0.05   |
> > > | FIF: free-form net, off manifold (fixed) | 0.04 ± 0.01   | 0.28 ± 0.03 | 0.54 + 0.03  | 0.60 ± 0.02   |
> > > | free-form net, on manifold (no fix)      | 19.54 ± 20.81 | 7.48 ± 5.40 | 29.03 ± 5.42 | 77.23 ± 16.55 |
> > >
> > > We therefore find that this fix was crucial for the successful training of free-form architectures as injective flows.

---

> > > > ### Comment · Reviewer_RfcP · 2023-11-22
> > > >
> > > > > The additional problem occurs regardless of whether the data is supported on a subspace or extends off-manifold. Consider figure 2. The pathological behavior illustrated on the left could still happen if the data distribution were a Gaussian with zero variance in the x direction, since the concentration effect due to curvature would still overwhelm the reconstruction error.
> > > >
> > > > Now I am confused again. Let's focus on the case where the data exactly populates a $d$-dimensional manifold in the $D$-dimensional data space, and assume that our model is expressive enough that we can exactly fit this manifold with the encoder and decoder (so we will have zero reconstruction error).
> > > >
> > > > Are you saying that in this scenario there is still a pathology that is different from the one pointed out by Brehmer & Cranmer? Can you elaborate a bit more on this?
> > > >
> > > > > In our experiments, this problem seems to be important
> > > >
> > > > Thanks for reporting these additional results. To understand what is going on, I would really appreciate seeing some reconstruction errors in addition to these FID-like metrics. Could you report them, please?

---

> > > > > ### Author Response · Authors · 2023-11-22
> > > > >
> > > > > Thank you for your questions, which we hope to answer below:
> > > > >
> > > > > > Are you saying that in this scenario there is still a pathology that is different from the one pointed out by Brehmer & Cranmer? Can you elaborate a bit more on this?
> > > > >
> > > > > Yes, exactly. Even if the data lies on a $d$-dimensional subspace and the model was rich enough to fit this manifold exactly, the on-manifold loss would prefer to learn a manifold that is strongly curved because this has a lower loss. The reason is that the reconstruction error is always bounded, but the negative log-likelihood with gradients on-manifold decreases arbitrarily.
> > > > >
> > > > > We have added Appendix F with Figures 12 and 13 showcasing exactly this case.
> > > > >
> > > > > > Could you report [reconstruction errors]?
> > > > >
> > > > > |              | Power        | Gas           | Hepmass       | Miniboone     |
> > > > > |--------------|--------------|---------------|---------------|---------------|
> > > > > | On manifold  | 237 ± 498    | 5835 ± 13006  | 119 ± 34      | 300 ± 160     |
> > > > > | *On manifold (outliers removed)* | *14 ± 27*      | *18 ± 31*       | *119 ± 34*      | *229 ± 23*      |
> > > > > | Off manifold | 0.072 ± 0.002 | 0.188 ± 0.012 | 2.569 ± 0.098 | 1.077 ± 0.011 |
> > > > >
> > > > > The reconstruction error is always much higher for on-manifold training compared to off-manifold, demonstrating the instability caused by on-manifold NLL evaluation in free-form networks. Note: the large standard deviations in on-manifold runs are typically the result of a single large outlier. We remove the largest outlier where applicable (“On Manifold (outliers removed)” row).

---

> > > > > > ### Comment · Reviewer_RfcP · 2023-11-23
> > > > > >
> > > > > > Thanks for the reco error results. These nicely fit into the story.
> > > > > >
> > > > > > However, I still don't follow how the issue you are describing (for the case where the data populates a $d$-dim. manifold) is different from the pathology pointed out in Brehmer & Cranmer. Like reviewer vmge, I think this discussion could use some more clarity.
> > > > > >
> > > > > > All in all, in its current form, I don't think this paper is quite up to the NeurIPS standard. But it clearly makes valuable points and I am not strongly against accepting it.

---

> > > > > > > ### Author Response · Authors · 2023-11-23
> > > > > > >
> > > > > > > We agree that the presentation of the pathologies and their fixes can be improved.
> > > > > > >
> > > > > > > Perhaps a better presentation is as follows: both of the pathologies we discuss are subcases of the pathology addressed by Brehmer and Cranmer. Both have the limiting case where data is projected to a single point on the manifold and the NLL is $-\infty$. We distinguish two cases of this pathology, which differ in which models they occur in and how they can be addressed:
> > > > > > >
> > > > > > > 1. When the model is inclined to learn the lowest entropy directions of the data, learning a manifold that does not align with the data manifold but intersects it, like in the linear example discussed by Brehmer and Cranmer. During joint learning, this pathology can also be fixed (to the best of our knowledge) by adding a sufficiently high reconstruction error.
> > > > > > > 2. When the model manifold concentrates the data by use of high curvature, thereby decreasing the entropy of the projection of the data to the manifold. This only occurs in nonlinear models, hence Brehmer and Cranmer did not notice this effect in their simple linear example. Importantly, this is **not** fixed by adding a reconstruction loss, as our toy examples in figs 2, 12 and 13 show. We fix it instead by eq. (18).
> > > > > > >
> > > > > > > Brehmer and Cranmer fix both pathologies by first learning the manifold via reconstruction loss, then learning the density in the latent space post hoc. When jointly learning the manifold and density on it, both have to be addressed simultaneously.

---

### Official Review · Reviewer_SnSS · 2023-10-30

**Soundness:** 4 excellent
**Presentation:** 4 excellent
**Contribution:** 3 good
**Rating:** 8
**Confidence:** 5

**Summary:**

This papers builds upon rectangular flows, a method for end-to-end training of injective normalizing flows. Three modifications are proposed: (a) not restricting the architecture with normalizing flows, since the reconstruction error encourages injectivity in it of itself, (b) a more efficient gradient estimator, thus addressing a main limitation of rectangular flows, and (c) a further modification to the gradient estimator, which changes the gradient itself and improves numerical stability.

Overall, the paper is well-written and I believe it makes a significant methodological contribution to the area of injective normalizing flows. That being said, I also believe that ablations are missing to properly identify the sources of empirical improvement that the authors observed.

------------------------------------------------------------------------------
11/22 UPDATE
------------------------------------------------------------------------------

The authors have adequately addressed the points I raised in my original review and I am thus increasing my score.

------------------------------------------------------------------------------
12/04 UPDATE
------------------------------------------------------------------------------

I did want to raise an additional point after having discussed the pathology described in section 4.2 with other reviewers. In particular, the bound in eq 14 is a lower bound on the loss, not an upper bound. Thus, if the entropy becomes arbitrarily negative, it does not automatically imply that the loss can become equally negative, as seems to be implied in the discussion following the equation ("the loss can continue to decrease without bound by reducing the entropy of the projected data").

While I do not believe the point made by the authors is wrong, I do believe that it requires additional arguments: for $\mathcal{L}_{NLL}$ to also become arbitrarily negative along with the entropy of projected data, the KL divergence in eq. 13 needs to not grow faster than the entropy. I still believe this can happen if the encoder and decoder have unrestricted architectures and collapse to point masses (since then the KL between projected data and the model would be controlled), but I am actually not sure if this can happen when the architectures are constrained to be injective. I also believe this might explain the results from the ablations, where the updated loss did not significantly improve results when using injective architectures.

If the paper is accepted, I would ask the authors to please flesh out the discussion around this, as I agree with reviewer RfcP that the current exposition in section 4.2 can lead to confusion.

**Strengths:**

This paper has several strengths:

1. It is well written and easy to follow, and I think the authors did a good job of deciding which material to include in the main manuscript and which details to include in the appendix.

2. It is well motivated, as I agree with the authors that the current injective flow literature uses overly restrictive architectures and/or computationally intensive training procedures.

3. The simple observation that, when the encoder $f$ is the left inverse of the decoder $g$, allows to write Jacobians of $g$ as Jacobians of $f$ is elegant, and does result in clear computational gains.

4. Empirical results are good, showing that the proposed method outperforms other injective flows and generative autoencoders.

**Weaknesses:**

5. In my view, the main weakness of the paper is the lack of ablations. As mentioned, the paper proposes 3 improvements over rectangular flows, and it is unclear how much each of these contributes to the empirical performance of the proposed method. I think table 1 provides a perfect test bed to carry out these ablations: results using the same architecture as rectangular flows should be added to the table, both (a) using the gradient estimator from eq 10, and (b) that from eq 16. Ideally, using eq 10 (and the same architecture as rectangular flows) would simply show a speed up and the same performance compared to rectangular flows, whereas using eq 16 should improve performance but not match the results of FIF with a fully flexible architecture. I would see this as strong empirical evidence backing up the claims in the paper. I will increase my score if these ablations are included.

6. While the authors include a discussion as why $x$ should be used (eq. 16) instead of $\hat{x}$ (eq. 10), I think there are several relevant points missing from the discussion: (a) why does the pathological behaviour described by the authors not happen in rectangular flows? Is it because the more restrictive architecture implicitly regularizes the curvature? Or is this actually a hidden issue in rectangular flows as well (the above ablation will obviously also help answer this question)?. (b) Since, when $f$ and $g$ are consistent, $f(x)=f(\hat{x})$, it seems to me like one can attempt to justify both objectives as attempting to maximize log-likelihood subject to perfect reconstructions. In this view, the problem of using $\hat{x}$ could be seen as an inappropriate way of enforcing the constraint through a penalty term. Could you further discuss? (c) There is also an additional computational benefit to using $x$ instead of $\hat{x}$, namely one less forward pass is required through the encoder, which I believe should also be mentioned.

Finally, some minor points:

- In the notation paragraph in sec 3, you write $f^{-1} = g$, which I think should be avoided: when $d<D$, $f$ cannot be an invertible function, since you defined its domain as $\mathbb{R}^D$ (its restriction to a manifold could of course be injective though, I am not saying there's anything fundamentally wrong here, just nitpicking the notation): I think it'd be better to stick to the language of left inverses.

- Missing period at the end of the injective flows paragraph in sec 3.

- Use \citep instead of \citet in the first paragraph of appendix E.3.

**Questions:**

7. As you point out in eq 1, injective flows typically have a low-dimensional flow $h$ on the latent space. One could also interpret this architecture as a flexible distribution $p_Z$ on latent space, given by $h$, along with a decoder $w \circ \texttt{pad}$; rather than thinking of their composition as the decoder $g$. Throughout the paper you mention making $g$ more expressive, but another interpretation is that you are making $w \circ \texttt{pad}$ more expressive, and reducing expressivity on the latent space (instead of a flow, you use a Gaussian or a mixture of Gaussians). Previous research has found benefits of having flexible distributions on latent space (rectangular flows prefer using a flow-based p_Z rather than fixing it as a Gaussian, and other works also recommend using flexible distributions on latent space, e.g. [1, 2]), is there a reason why you do not use more flexible $p_Z$?

[1] Diagnosing and Enhancing VAE Models, Dai & Wipf, ICLR 2019

[2] Diagnosing and Fixing Manifold Overfitting in Deep Generative Models, Loaiza-Ganem et al., 2022

---

> ### Author Response · Authors · 2023-11-15
>
> Thank you for your thorough and helpful feedback.
>
> ### Ablation comparing estimators
>
> We are still working on implementing the ablation and will come back with results before the end of the rebuttal.
>
>
> ### Why $x$ instead of $\hat x$?
>
> **No pathological behavior in rectangular flows?**
>
> From a theoretic point of view, rectangular flows should also suffer from pathologically strong curvature since their conjugate gradient does evaluate the likelihood term on-manifold.
>
>
> **Using $\hat x$ is an inappropriate way to enforce consistency**
>
> It is true that if $f$ and $g$ are consistent then $f(x) = f(\hat x)$, but this is not true for the Jacobians: $f’(x) \neq f’(\hat x)$ in general (see fig. 4 and section B.2 in the appendix for a discussion). Hence substituting $x$ with $\hat x$ is not just another way of enforcing consistency, it gives a different numerical result even in consistent models, and helps fight against degenerate high-curvature solutions.
>
> **Computational benefit**
>
> You are correct that there is a computational benefit to using $x$ instead of $\hat x$ (one less forward pass). We will add a mention of this in section 4.2.
>
> ### More flexible p(z)
>
> We do find it beneficial in practice to have a few residual blocks which work with the latent dimension. This can be interpreted as being the same structure of an injective flow, which has a latent space flow. Hence, you can view our model as having an expressive latent distribution.
>
> We hope that this addresses your questions and we look forward to further discussion.

---

> > ### Comment · Reviewer_SnSS · 2023-11-20
> >
> > Thank you for your reply, I am happy with your clarifications and look forward to seeing the ablations.

---

> ### Author Response · Authors · 2023-11-20
>
> # Ablation results:
>
> We perform the requested ablations to compare the on- and off-manifold variants of our loss using a free-form and an injective flow architecture. The injective flow experiments were performed with both FIF and RF hyperparameters. The results, along with the original results from our submission (top two rows):
>
> | Experiment                               | power         | gas         | hepmass      | miniboone     |
> |------------------------------------------|---------------|-------------|--------------|---------------|
> | RF: injective flow via RF loss with CG   | 0.08 ± 0.02   | 0.11 ± 0.02 | 0.78 ± 0.19  | 1.00 ± 0.05   |
> | FIF: free-form net, off manifold         | 0.04 ± 0.01   | 0.28 ± 0.03 | 0.54 + 0.03  | 0.60 ± 0.02   |
> |------------------------------------------|---------------|-------------|--------------|---------------|
> | free-form net, on manifold               | 19.54 ± 20.81 | 7.48 ± 5.40 | 29.03 ± 5.42 | 77.23 ± 16.55 |
> | injective flow, off manifold, FIF hparams| **0.11 ± 0.06**   | **0.45 ± 0.09** | **1.30 ± 0.14**  | **1.55 ± 0.04**   |
> | injective flow, off manifold, RF hparams | 0.98 ± 0.69   | 6.16 ± 4.20 | 2.02 ± 0.74  | 1.80 ± 0.10   |
> | injective flow, on manifold, FIF hparams | 3.71 ± 2.19   | 0.40 ± 0.22 | **0.71 ± 0.05**  | 3.13 ± 0.42   |
> | injective flow, on manifold, RF hparams  | **0.33 ± 0.22**   | **0.33 ± 0.17** | 0.82 ± 0.07  | **1.84 ± 0.11**   |
>
> Note: bold entries show the best hyperparameters (out of FIF or RF) for each experiment.
>
> ## Conclusion:
>
> The ablation shows that (a) the free-form network is unstable for the on-manifold variant of our loss, and (b) the injective flow is more stable against evaluating our loss on manifold:
> For the injective architecture, on- and off-manifold settings are both stable, indicating that there is a stabilizing inductive bias of the architecture. Note that they do not perform as well as the original networks, but this is probably due to the lack of tuning hyperparameters.
>
> ## Details:
> - FIF hparams means the same training hyperparameters as in our paper
> - RF hparams means the same training hyperparameters (including a warm up in the NLL loss) as detailed in the appendix of the rectangular flows paper, except that a fixed number of epochs (150) was used rather than using early stopping
> - Injective flow means the RNVP-based architecture detailed in the appendix of RFs
> - Hepmass is trained with LR=1e-4 instead of LR=3e-4 when using FIF hparams
> - Injective flow with FIF hparams have 4 instead of 5 runs each

---

> > ### Comment · Reviewer_SnSS · 2023-11-20
> >
> > Thank you for posting these ablations. I do have some follow-up questions/comments:
> >
> > 1. I completely agree that these results show the numerical instabilities that you mentioned on the paper when using free-form architectures. I also believe the point that injective flows are more robust to on/off-manifold evaluation is interesting and should be mentioned somewhere.
> >
> > 2. You mention that injective flows "do not perform as well as the original networks, but this is probably due to the lack of tuning hyperparameters". I don't completely understand this, specially for the RF hyperparameter settings. I believe the rows "injective flow, on manifold, RF hyperparams" and "RF: injective flow via RF loss with CG" should behave very similarly, as they should just be computing the same gradients in different ways (and also using the same hyperparameter). Am I misunderstanding something? Could you comment of the observed differences between these rows?
> >
> > 3. It does not appear that you get consistent improvements by using the off manifold formula when restricting the architecture to injective flows (i.e. the "injective flow, off manifold, FIF hparams" row is not consistently better than the "injective flow, on manifold, FIF hparams" row; and similarly, the "injective flow, off manifold, RF hparams" row is not consistently better than the "injective flow, on manifold, RF hparams"). This seems to be in contradiction to the narrative that off manifold is always better, could you comment on this?

---

> > > ### Author Response · Authors · 2023-11-21
> > >
> > > Thanks for considering our changes and following up on your questions!
> > >
> > > 1. We would of course include this additional information by expanding Table 1 as you suggested and make a comment on it in the text. Thank you for suggesting this ablation study.
> > > 2. It is indeed surprising how differently the injective flow behaves when using the CG loss and our on-manifold loss. We have three hypotheses for the source of the difference:
> > >     1. The losses only give the same gradients if the reconstruction loss is minimal. While $f’(\hat x) g’(f(x)) = I$ always holds for injective flows, $g’(f(x))^\dagger = (g’^T g’)^{-1} g’^T = f’(\hat x)$ requires additional constraints (simple example with 1d latent space: $f(x) = a^T x, g(z) = bz$ with $a^T b = 1$ but $a \neq b^\dagger$, e.g. $a = (1, 1), b = (1, 0)$). These constraints are fulfilled when the reconstruction loss is minimal. Unfortunately this point is not explored in our paper, we will update the theoretical section accordingly. We observe that reconstruction loss continues to decrease during the training, suggesting that it is never minimal, which could be the source of the difference.
> > >     2. Our loss sends gradient from the log det to the encoder, whereas the RF loss sends it to the decoder. While this shouldn’t make a difference since encoder and decoder in an injective flow share weights, perhaps there is a difference in practice we are not aware of.
> > >     3. The numbers reported with the CG loss are from the rectangular flows paper, using their code. The numbers when using our loss ("injective flow, on manifold, RF hyperparams") are our reimplementation of the architecture and training, using our training code (due to the difficulty of adapting the training framework of the RF codebase on short notice). While we took care in faithfully reproducing all the details, including comparing against the RF code, it’s possible there are still differences.
> > > 3. When using the injective flow architecture, neither the on- nor off-manifold loss shows a clear advantage. We speculate that due to some helpful inductive bias of the injective flow, it is not susceptible to pathological curvature effects. However, we have yet to identify the source of this inductive bias. On the other hand, the off-manifold loss is clearly better for free-form injective flows. Since the free-form variant is faster to train and ultimately results in better performance, this highlights the importance of the off-manifold loss.
> > >
> > > If you have any further questions, we are happy to provide answers during the remaining time of the discussion.

---

> > > > ### Comment · Reviewer_SnSS · 2023-11-21
> > > >
> > > > Thank you for your answers. I do not have any additional questions, but I did want to mention that the results of the ablations are quite surprising: it really seems like the proposed improvements over rectangular flows really become effective when using free-form architectures, rather than when using injective flows. While I agree that the best results overall being obtained with free-form architectures highlights the relevance of the proposed improvements, I do think the ablations show that the empirical improvements come from the combination of free-form architectures with the other methodological changes, rather than every single change being an improvement in it of itself, which is the current way the paper reads.
> > > >
> > > > Nonetheless I still believe this is a strong paper that improves upon existing injective flows. Thus, as long as the authors promise to include these results and the corresponding discussions in the paper, I will increase my score as I stated in my original review.

---

> > > > > ### Author Response · Authors · 2023-11-22
> > > > >
> > > > > We agree with your interpretation of the ablation. We have included it into a preliminary revised version of the paper (see new uploaded PDF).
> > > > >
> > > > > We thank the reviewer for the constructive and engaged discussion.

---

### Official Review · Reviewer_vmge · 2023-10-31

**Soundness:** 2 fair
**Presentation:** 2 fair
**Contribution:** 3 good
**Rating:** 5
**Confidence:** 4

**Summary:**

A new Injective Flow named free-form injective ﬂow (FIF) is proposed. FIF is developed based on the rectangular flow (Caterini et al., 2021) and change of variables across dimensions. Different from the rectangular flow, FIF leverages the auto-coding architecture to approximately but efficiently calculate the gradient of maximum likelihood surragate wrt the parameters. The authors also identify pathological behavior in the naive application of maximum likelihood training and propose a fix.

**Strengths:**

The presented techniques are original, with contributions on removing limitations from prior methods.

The presented techniques are interesting and potentially valuable.

**Weaknesses:**

The clarity should be significantly improved. For example, many important derivations should be moved to the main manuscript, and important assumptions should be highlighted.

**Questions:**

Without architectural constraints, how to guarantee that $det [g′(z)^T g′(z)] > 0$?

In the paragraph before Eq. (12), what are the assumptions underlying $f(\hat x)=f(x)$? Also, why does Eq. (12) hold true? If $p_{data}(x)=\hat p_{data}(\hat x)$, then the right-hand side of Eq. (14) is fixed, right?

After adopting the modification in Eq. (16), the final objective in Eq. (18) (or its first two terms) ultimately is not identical to the negative maximum likelihood, right? If so, what are the differences?

---

> ### Author Response · Authors · 2023-11-15
>
> Thank you for your feedback. We are happy to provide the additional details:
>
> ### How to guarantee a positive determinant?
>
> The reconstruction loss is the only direct loss on the decoder. Reconstruction loss cannot be minimized unless the decoder is injective. Therefore if the reconstruction loss is well-optimized, the determinant is very likely (although not guaranteed) to be positive.
>
> ### Assumption underlying $f(\hat x) = f(x)$
>
> By definition, $\hat x = g(f(x))$. In the section you are referring to (4.2) we are assuming that $f$ and $g$ are pseudoinverses for the purpose of the derivation, then later drop that assumption. Under the assumption, $f(\hat x) = f(x)$. Sorry for the confusion, we will revise the text to make this assumption more clear.
>
> ### Eq. (12)
>
> To clarify your confusion: it is not the case that $p_\text{data}(x) = \hat p_\text{data}(\hat x)$. We replace $p_\text{data}$ with $\hat p_\text{data}$ in eq. (12) because the quantity in the expectation is invariant to projection, so we can replace the original data distribution with the projected distribution.
>
> ### Relation to maximum likelihood (ML)
>
> You are right that the final objective is not exactly maximum likelihood. The differences are:
> - We introduce a reconstruction loss which ensures that high-entropy directions are encoded in the latent space (they would be ignored with a ML loss only).
> - We evaluate the encoder Jacobian off-manifold to ensure that the solution does not become degenerate due to excessive curvature (see section 4.2 “Towards a well-behaved loss”).
> Given that the two modifications make joint maximum likelihood and manifold learning tractable, we argue that it is justified to denote this as maximum likelihood training. Note that maximum likelihood training alone would not lead to sensible solutions when using bottleneck architectures, so our modifications are necessary to learn ML and manifold simultaneously.
>
> We hope that this answers your questions and we look forward to further discussion.

---

> > ### Comment · Reviewer_vmge · 2023-11-22
> >
> > Thank you for your reply, which addressed some of my concerns. I have three follow-up questions.
> >
> > I now understand that the derivations in Section 4.2 are based on the assumption that $f$ and $g$ are pseudoinverses for all $x \sim p_{data}(x)$.
> >
> > (1) How does that assumption affect Eq. (14)? Is it possible that the NLL or the entropy decreases without a bound, based on that assumption?
> >
> > (2) How did you guarantee that assumption in the proposed loss in Eq. (18)?
> >
> > Another question is associated with the core advantage of a flow -- the exact evaluation of the likelihood.
> >
> > (3) The proposed FIF gives up that core advantage with Eq. (18), for what? This should be discussed in detail.

---

> > > ### Author Response · Authors · 2023-11-22
> > >
> > > Thank you for your continued interest and questions, here are our answers:
> > >
> > > 1. Yes, it is possible that the NLL decreases without bound, even if $f$ and $g$ are pseudoinverses. In fact, in figure 2, we use an $f$ such that $g \circ f$ is a projection to the closest point on the learned manifold. This guarantees that $f$ is pseudoinverse to $g$ since it means that $f \circ g$ is the identity. Even when this is the case, the pathological behavior occurs.
> > > 2. We do not guarantee that $f$ and $g$ are pseudoinverse in eq. (18). Instead we use the reconstruction loss as a soft constraint such that $f$ and $g$ are approximately pseudoinverse. We set the reconstruction loss high enough such that this is a safe assumption.
> > > 3. Eq. (18) lets us train free-form networks as injective flows, something which was not possible prior to this work. It is not possible in general to have cheap likelihoods when using free-form networks as transformation functions. Nevertheless, free-form networks are more flexible than restrictive ones and, in our experience, require fewer computational resources to train. In addition, we experienced no trouble in practice calculating likelihoods at inference time. It requires the calculation of the full Jacobian matrix, but this was less costly than we expected. Crucially, we don’t ever have to calculate the full Jacobian during training.

---

> > > > ### Comment · Reviewer_vmge · 2023-11-23
> > > >
> > > > Sorry, but I am confused again.
> > > >
> > > > The change of variables across dimensions in Eq. (2) is only valid iff $f\circ g(z)=z$ and $g\circ f(x)=x$.
> > > >
> > > > However, in Section 4.2, you only assumed $f\circ g(z)=z$, right? Based on $f\circ g(z)=z$, you derived Eqs. (12) and (14) and drew the conclusion following Eq. (14), where the RHS could decrease without bound because $g\circ f(x)=x$ may not be true. Otherwise, if both $f\circ g(z)=z$ and $g\circ f(x)=x$ are fulfilled, then $p_{data}(x)=\hat p_{data}(\hat x)$.
> > > > So it's an auto-encoder with $f\circ g(z)=z$ that is discussed in Section 4.2, right?
> > > >
> > > > Overall, I don't think the discussions here are clear. There are two fundamental problems, one of which is associated with training an injective flow (on a dense data distribution), while the other is related to maximum likelihood learning on a manifold (clearly a pathological learning problem). It seems both problems are mixed up.
> > > >
> > > > Could the authors elaborate on this? Actually, I am quite curious about how the proposed fix in Eq. (18) helps stabilize the maximum-likelihood-like learning on a manifold.

---

> > > > > ### Author Response · Authors · 2023-11-23
> > > > >
> > > > > Thank you for your continued interest. To be sure we address all your concerns adequately, we list several and provide an answer to each. This may also be of interest to the other reviewers.
> > > > >
> > > > > **Concern**: the different problems with joint maximum likelihood and manifold learning are mixed up.
> > > > >
> > > > > See our comment to reviewer RfcP (recopied here for convenience):
> > > > >
> > > > > We agree that the presentation of the pathologies and their fixes can be improved.
> > > > >
> > > > > Perhaps a better presentation is as follows: both of the pathologies we discuss are subcases of the pathology addressed by Brehmer and Cranmer. Both have the limiting case where data is projected to a single point on the manifold and the NLL is $-\infty$. We distinguish two cases of this pathology, which differ in which models they occur in and how they can be addressed:
> > > > >
> > > > > 1. When the model is inclined to learn the lowest entropy directions of the data, learning a manifold that does not align with the data manifold but intersects it, like in the linear example discussed by Brehmer and Cranmer. During joint learning, this pathology can also be fixed (to the best of our knowledge) by adding a sufficiently high reconstruction error.
> > > > > 2. When the model manifold concentrates the data by use of high curvature, thereby decreasing the entropy of the projection of the data to the manifold. This only occurs in nonlinear models, hence Brehmer and Cranmer did not notice this effect in their simple linear example. Importantly, this is **not** fixed by adding a reconstruction loss, as our toy examples in figs 2, 12 and 13 show. We fix it instead by eq. (18).
> > > > >
> > > > > Brehmer and Cranmer fix both pathologies by first learning the manifold via reconstruction loss, then learning the density in the latent space post hoc. When jointly learning the manifold and density on it, both have to be addressed simultaneously.
> > > > >
> > > > > **Concern**: if the data lies on a $d$-dimensional manifold, the pathologies discussed don’t apply.
> > > > >
> > > > > The pathologies still apply. Let’s refer to the new figures 12 and 13 in the appendix. They show cases where the data and latent manifolds have the same dimension but do not overlap exactly. We show that this toy model will always prefer high-curvature solutions, no matter how high the reconstruction loss is. The data can be arbitrarily concentrated in the projection to the model manifold when curvature is high, thus reducing the NLL towards negative infinity. Importantly, this pathology cannot be fixed by a reconstruction loss. We address this by evaluating the encoder Jacobian off-manifold (see eq. (16)), motivating eq. (18).
> > > > >
> > > > > **Concern**: How does evaluating the encoder Jacobian off-manifold fix the pathology of spuriously curved manifolds?
> > > > >
> > > > > We refer the reader to Appendix B.2. One way to interpret this is that the on-manifold Jacobian does not correct for how $x$ is stretched or compressed due to the curvature of the manifold, but the off-manifold Jacobian does. We note that this a well-motivated heuristic fix at the current stage, and further research is required for stronger theoretical justification and potential further improvements.
> > > > >
> > > > > **Concern**: Why is the off-manifold behavior of a bottleneck model important?
> > > > >
> > > > > There are at least three situations where the behavior outside of the manifold is important: :
> > > > > - Most real data are not exactly on the manifold due to noise and other unavoidable perturbations.
> > > > > - In the midst of training, the model has not yet converged to the correct manifold.
> > > > > - Real data may exhibit slight distribution shifts from one context to the next, and we would like the model to be robust against this.
> > > > >
> > > > > In all three cases, it matters a great deal exactly how off-manifold data is projected onto the manifold.

---

### Official Review · Reviewer_KrZV · 2023-10-31

**Soundness:** 3 good
**Presentation:** 4 excellent
**Contribution:** 3 good
**Rating:** 8
**Confidence:** 5

**Summary:**

This paper proposes a new technique for training flow-based architectures with manifold structure, entitled the "Free-form Injective Flow". This technique is derived by loosening the architectural constraints on previously-proposed injective flows; in particular, the proposed autoencoder is completely unconstrained besides using a pre-specified latent dimensionality $d$. Issues with the loss used for previous injective flows are also identified, and this paper derives a novel loss function to address those issues. This loss is computationally tractable while maintaining stability. Experiments are performed to compare with previous injective flow techniques, and other types of autoencoders.

**Strengths:**

I'll enumerate the strengths below for ease of reference in discussion. These are not listed in order of importance.

1. The paper is generally very well-written and comes up quite polished. I'll outline below:
    - The introduction is very clean. The motivation for the method is clearly laid out.
    - The paper is well-situated amongst the related work.
    - The background is easy to digest.
    - The path to the final model in Section 4 is laid out well.
    - The appendix is quite thorough
2. The autograd / linear-algebra-type derivations were quite well-done. I have personally always appreciated works that don't just blindly apply basic automatic differentiation techniques and instead think deeper about the problem and the requisite gradient estimators.
3. I like the use of figures. Figure 2 in particular is quite nice for explaining how the technique improves on previous injective flow-based methods. Figure 3 also clearly demonstrates the trade-off between reconstruction and log-likelihood.
4. I feel well-convinced that this technique is clearly better than previous injective flow techniques, both in terms of representation power and computational tractability.

**Weaknesses:**

I'll write weaknesses in a list as well. Again this list is not ordered in terms of importance.

1. In the end, this paper could be summarized as simply training autoencoders with a different training loss, with the loss motivated by previous work in injective flows. The novelty and significance of this particular choice of loss over other types of autoencoder losses is not completely clear for a couple of reasons: (i) Table 3 is not convincing, as the best results are still produced by other autoencoders, and (ii) more modern autoencoder architectures are not compared against. Furthermore, this paper does not necessarily maintain all of the benefits of injective flows - mainly, we do not get exact inverses on the projections to the learned manifold. To summarize, I think *some* degree of discussion is warranted on the benefits of using this approach over other generative autoencoders, as the benefits over other injective flows are comparatively very well-documented here.
2. This paper is missing a dedicated limitations section. This is partially covered by the conclusion, but not completely, and would show some more perspective from the authors considering the weaknesses laid out here.
3. In section 5.3, it is suggested that Inception Scores are a reliable measure of diversity, although I don't know if that's actually a modern viewpoint. Furthermore, the Inception Scores generally seem just in-line with other methods, or worse at times. I am also confused about why the best Inception Scores are not bolded in Table 3.
4. It seems like Section 5.2 and 5.3 are out-of-order on how things are defined. For example, the FID acronym is both cited and defined in 5.3, yet referred to in 5.2. Table 2 also requires more of a description -- including what "IS" is, and what the two samplers are -- some of which is contained within the caption of Table 3. I would just suggest making the requisite definitions in Section 5.2 first and then using acronyms or reduced descriptions in Section 5.3 as appropriate.
5. It is discussed twice that traces are performed in the order $f' g'$, and that details are in the Appendix -- however, there is certainly space in the paper to provide a bit more discussion on that.
6. The paragraph on Page 6 starting with "Unfortunately" is not sufficiently convincing: I don't think Fig 2 proves that reconstruction error is insufficient, as you could imagine that reconstruction becomes increasingly more difficult if the entropy becomes negative infinity which should therefore regularize the solution on the left to some extent so that it does not fall on the degenerate, negative-infinite entropy solution.
7. It is suggested that rectangular flows require $O(d)$ `vjp`s / `jvp`s for convergence, but practically conjugate gradient has exponential convergence and thus much fewer iterations suffice.
8. It is stated that the surrogate loss is only accurate if $f$ and $g$ are optimal with respect to reconstruction, and then assumed that this is indeed fulfilled by optimizing the reconstruction loss and by the fact that training is stable. However, I don't think this is fully proven:
    - The reconstruction error is not checked in the paper
    - One of the other changes to the loss function may be responsible for the training stability
    - Trade-offs between optimizing the reconstruction error and reconstruction the likelihood contribution may prevent the reconstruction loss from being fully optimized (cf. Figure 3)
9. There are no samples provided for the generative methods, which suggests that the generation quality may not actually be that good. FID has recently come under more scrutiny as an evaluation metric and so supporting the FID numbers with actual generated samples would be useful.

**Questions:**

1. Why is the training time speedup inconsistent in Table 1? It doesn't seem to scale with dimension in any predictable way.
2. What is the definition of entropy as in e.g. (13) for a distribution that is supported on a manifold?

**POST-REBUTTAL**

I'll be upgrading my score after discussion with the authors.

---

> ### Author Response · Authors · 2023-11-15
>
> Thank you for your constructive and detailed feedback. Let us address the weaknesses you point out:
>
> 1. To clarify the advantages of our model compared to the VAE and its variants: They are limited by (i) their variational family, and (ii) by learning a distribution in the embedding space, which may be limiting if the data actually lies on a manifold (cf. manifold overfitting: Dai and Wipf, 2019 and Loaiza-Ganem et al. 2022). While (i) can be addressed by more expressive likelihoods/posteriors, this comes with additional hyperparameters. Our model only has the architecture and beta as hyperparameters.
> 2. Adding a limitations section is a good idea which we will add for the camera ready version. We will address: The Pythae benchmark is limited because it does not scale training to competitive image generation, the FID and IS metrics are limited, and we observed that architectural changes can significantly improve results. Also, while the empirical results are good, theoretically showing whether the final loss actually learns the exact distribution on the manifold is still open; in particular when the reconstruction loss is not at optimality. Finally, the reconstruction weight is left open as a hyperparameter to tune for each new data set.
> 3. We included Inception Scores because they are part of the Pythae benchmark. We included the Inception scores because they indicate diversity. We do not bold them since FID is more suited at comparing sample quality to real data.
> 4. Thank you for this suggestion. We agree that some definitions are out of order and it makes sense to make all the definitions in section 5.2.
> 5. We can add more detail about the $f’ g’$ order in the trace, since this is crucial in order to reduce variance.
> 6. To elaborate on why reconstruction loss alone is not sufficient: imagine in fig. 2 the limit where the learned manifold has infinitely high curvature. In this case, all data will be projected to the same point, so the reconstruction error will take on some finite value. In contrast, the NLL will be negative infinity. Hence we can see the problem: this solution has an infinitely small loss, for any (finite) $\beta$ we choose. Therefore reconstruction error alone is insufficient.
> 7. You are right that the convergence speed is exponential in the number of iterations, but the convergence behavior might be different with dimension. We will weaken the wording in the introduction and point to our experiment, where we find that wall clock time is longer with CG than with our method.
> 8. We always monitored reconstruction loss on validation data during training and found that it was always low in our best-performing models. For example, Figure 1 shows reconstructions of CelebA validation data. Regarding training, we find that when reconstruction loss is high (due to too low $\beta$) training becomes unstable. We do find that reconstruction loss never goes to zero, which it can’t given the abundance of details in face images.
> 9. The aim of the Pythae benchmark is to compare models on a given computational budget, and the evaluation metrics are FID and IS. We agree that there are flaws to these metrics, but they are currently the only feasible way to compare against a large number of competing generative autoencoders. See Figure 1 for samples from our improved architecture. See Figure 11 in the Pythae paper for samples from competitor models: https://arxiv.org/pdf/2206.08309.pdf
>
> Finally, to address your additional questions:
>
> 1. The inconsistency in training time in table 1 is due to the unpredictability of rectangular flow training, which terminates when no further improvement is made. Training time can vary widely between runs. Please see table 5 on page 30 in the appendix for a fuller picture.
> 2. If we restrict integration to the manifold, the entropy has a sensible definition. Suppose that $\hat p$ is normalized on the manifold: $\int \hat p(\hat x) d \hat x = 1$ where integration is only over the manifold. Then the entropy of $\hat p$ is $-\int \hat p(\hat x) \log \hat p(\hat x) d \hat x$, again integrating only over the manifold. Note that the KL divergence in eq. (13) is defined similarly, with integration restricted to the manifold. There is a typo in eq. (13): the expectation should be over $\hat p_\text{data}(\hat x)$, not $p_\text{data}(\hat x)$.
>
> We hope that this clarifies our view on the weaknesses you mention, and are happy to discuss these or additional points in the discussion.
>
> **References**
>
> - Dai and Wipf. Diagnosing and enhancing VAE models. ICLR, 2019.
> - Loaiza-Ganem et al. Diagnosing and Fixing Manifold Overfitting in Deep Generative Models. TMLR, 2022.

---

> > ### Comment · Reviewer_KrZV · 2023-11-20
> >
> > Thanks for your reply! I have some follow-up questions - for numbers I leave blank I have no follow-up and accept your response.
> >
> > 1. Your response focuses on a comparison to VAE and its variants - but how about non-variational autoencoders? A standard, modern/highly-performant autoencoder does not necessarily fall victim to the limitations you noted for VAEs, including manifold overfitting. I am still curious about how you situate your work against more standard autoencoders.
> > 2. Thank you for the discussion on limitations.
> > 3. Again "We included the Inception scores because they indicate diversity" is stated, but with what backing? And is this truly still a valid viewpoint considering the advancements made to attempt to quantify diversity of generated datasets?
> > 4.
> > 5.
> > 6. I see, that makes more sense. Is it worth including a sentence on this in the main paper?
> > 7.
> > 8.
> > 9. I apologize as I forgot about the samples early on in the text. Can more samples be included in the experiments section or the appendix?

---

> > > ### Author Response · Authors · 2023-11-21
> > >
> > > Thanks for your follow up questions! We address these below:
> > >
> > > > Your response focuses on a comparison to VAE and its variants - but how about non-variational autoencoders?
> > >
> > > Here are some points comparing our model to non-VAE autoencoders included in the benchmark:
> > > - Vanilla autoencoders: not generative models; our model can generate samples
> > > - Adversarial AE and VAEGAN: require an additional adversarial model and potentially unstable adversarial training and mode collapse; our model requires only encoder and decoder, training is based on maximum likelihood so no mode collapse
> > > - Regularized AE: requires two-step training to learn the latent distribution; our model uses a standard normal latent and is trained in one step
> > > - Wasserstein AEs: Regularize the latent space via MMD and not ML. MMD suffers from the curse of dimensionality as can be seen by its diminishing performance going from MNIST to CIFAR.
> > >
> > > Are there any other specific autoencoder models you have in mind for a comparison?
> > >
> > > > And is this [Inception scores indicate diversity] truly still a valid viewpoint considering the advancements made to attempt to quantify diversity of generated datasets?
> > >
> > > We base this viewpoint on a subjective comparison of the generated CelebA images against Inception scores for the models in the Pythae benchmark, see fig. 11 and table 7 in [1]. For example an IS of 1 (the min possible value) corresponds to a single blank image being generated and an IS of 2.8 (RAE-L2) corresponds to very diverse images. We are very interested in metrics with better performance and would greatly appreciate some pointers to relevant literature.
> > >
> > > > Is it worth including a sentence on this [why reconstruction is not enough] in the main paper?
> > >
> > > Certainly. We will add this to the discussion in section 4.2.
> > >
> > > > Can more samples be included in the experiments section or the appendix?
> > >
> > > We have updated our submission pdf by adding a comparison of samples from all models considered in the Pythae benchmark, see Figure 11 at the end of the appendix.
> > >
> > > [1] Chadebec et al., Pythae: Unifying Generative Autoencoders in Python A Benchmarking Use Case, NeurIPS 2022

---

> > > > ### Comment · Reviewer_KrZV · 2023-11-23
> > > >
> > > > Thanks for the more thorough treatment of autoencoders! I'm quite happy with the discussion there and I think it would be worthwhile to include in the paper, as that was the biggest weakness to me.
> > > >
> > > > As for diversity, some popular and more recent metrics are [Recall](https://arxiv.org/abs/1904.06991), [Coverage](https://arxiv.org/abs/2002.09797), and [Vendi Score](https://arxiv.org/abs/2210.02410), along with variants. Some additional perspective and a summary of performance is provided [here](https://arxiv.org/abs/2306.04675). I believe some of these metrics can be easily computed and might give more insight into the overall performance of the method.
> > > >
> > > > Thank you for responding to my other concerns. Altogether, I'll raise my score to an 8 here, under the assumption that the authors include some of the additional discussion that we've talked about.

---

### Author Response · Authors · 2023-11-23

We summarize the discussion to the following main changes to the manuscript:

- Fix the order of definitions in sections 5.2 and 5.3
- Add more detail about the order of the trace $f'g'$
- Weaken the wording about CG's convergence speed in the introduction
- Revise the text in section 4.2 to make the assumption of $g$ and $f$ being pseudoinverses clear
- Mention the computational benefit of the off-manifold formulation in 4.2
- Mention limitations about FID and IS as metrics
- Report reconstruction errors
- Separate the description of the subclasses of pathologies arising from the NLL loss on the projected data (Brehmer & Cranmer linear pathology + new nonlinear pathology)
    - Add example for new pathology occurring on data known to live on manifold -> Done in Appendix F
- Expand Table 1 with the ablation study -> Done in Section 5.1
- Add a comparison of samples from Pythae benchmark models and FIF to the appendix -> Done in Appendix E

We thank all reviewers for the vivid discussion period and the extensive interest in our work.

---

### Meta-Review · Area_Chair_AASN · 2023-12-10

**Metareview:**

The paper suggests an improved way to optimize NLL with injective flows. This is a well written paper with a clean contribution moving to a more general architectures for injective flows, and a nice usage of structure to improve computation of jacobians (Jacobian of the encoder as approximation for the "inverse" Jacobian of the decoder). The main criticism revolved around clarity of writing, usefulness of injective flows, and the fact that the concept of optimizing manifold and flow simultaneously is not new, and the quality of results. While during reviewers' discussion period two reviewers supported the paper, the other two did not strongly object to accepting the paper but acknowledged some pathologies when training the likelihood when only the reconstruction loss is used to make sure encoder is the inverse of decoder over its image.

**Justification For Why Not Higher Score:**

While two reviewers supported this paper during the discussion period, the other two were reluctant due to several factors including potential pathologies not treated/discussed sufficiently by the authors, usefulness of the method and the quality of the experimental section. See more details in the Meta Review.

**Justification For Why Not Lower Score:**

All reviewers agreed this paper offers an interesting contribution for injective flows. See more details in the Meta Review.

---

### Decision · Program_Chairs · 2024-01-16

Accept (poster)